# Liquid-liquid phase separation and extracellular multivalent interactions in the tale of galectin-3

Yi-Ping Chiu[1,4], Yung-Chen Sun[1,4], De-Chen Qiu[1], Yu-Hao Lin[1], Yin-Quan Chen[2], Jean-Cheng Kuo [1,2] & Jie-rong Huang [1,3✉]

Liquid-liquid phase separation (LLPS) explains many intracellular activities, but its role in extracellular functions has not been studied to the same extent. Here we report how LLPS mediates the extracellular function of galectin-3, the only monomeric member of the galectin family. The mechanism through which galectin-3 agglutinates (acting as a "bridge" to aggregate glycosylated molecules) is largely unknown. Our data show that its N-terminal domain (NTD) undergoes LLPS driven by interactions between its aromatic residues (two tryptophans and 10 tyrosines). Our lipopolysaccharide (LPS) micelle model shows that the NTDs form multiple weak interactions to other galectin-3 and then aggregate LPS micelles. Aggregation is reversed when interactions between the LPS and the carbohydrate recognition domains are blocked by lactose. The proposed mechanism explains many of galectin-3's functions and suggests that the aromatic residues in the NTD are interesting drug design targets.

[1] Institute of Biochemistry and Molecular Biology, National Yang-Ming University, No. 155 Section 2 Li-nong Street, Taipei 11221, Taiwan. [2] Cancer Progression Research Center, National Yang-Ming University, No. 155 Section 2 Li-nong Street, Taipei 11221, Taiwan. [3] Institute of Biomedical Informatics, National Yang-Ming University, No. 155 Section 2 Li-nong Street, Taipei 11221, Taiwan. [4]These authors contributed equally: Yi-Ping Chiu, Yung-Chen Sun. ✉email: jierongh@ym.edu.tw

How molecules "know" where and when to react in the presence of multiple simultaneous cellular reactions is an open question. Lipid-bound membranes act as compartments that separate reactions (e.g., the synthesis of ATP inside mitochondria; the degradation of proteins inside lysosomes)[1], allowing them to progress efficiently. Another mechanism through which biological activities are regulated in cells is the formation of biomolecular condensates (also known as membraneless organelles), such as stress granules, nucleoli, and Cajal bodies[2]. Although these condensates have been observed for decades, their biophysical properties have only recently been investigated[2–4]. The formation of biomolecular condensates is governed by multivalent proteins that undergo LLPS[5]. RNA binding proteins such as hnRNP A1, hnRNP A2, FUS, and TDP-43 are some of the most studied examples. The multivalency of these proteins is typically the result of repeated amino-acid motifs in intrinsically disordered regions[6], responsible for reversible aggregation and the formation of stress or RNA granules[7–10]. On the other hand, dysregulated condensate formation, because of point mutations[9,10] or post-translational modifications[11,12], can lead to pathological fibrilization. LLPS is also involved in DNA transcription regulation, notably in the functions of the disordered C-terminal domain of RNA-polymerase II and associated proteins[13,14], the disordered activation domains of transcription factors[15–17], and heterochromatin proteins[18,19]. Other cellular functions also known to be mediated by LLPS include cell cycle control[20,21], synaptic transmission[22,23], autophagic degradation[24], cancer suppression[25], nuclear pore passage[26,27], protein quality control[28], and signal transduction[29,30]. The question remains however whether LLPS, or the underlying mechanisms, also mediate functions in the extracellular milieu.

Cell-cell adhesion and extracellular signal response are reminiscent of granule formation and signal transduction inside cells. Lectins, which selectively bind to carbohydrates, are involved in many important extracellular functions such as cell agglutination and immune response[31]. Most lectins are membrane proteins, partially anchored to the cell membrane with outward carbohydrate-recognition domains (CRDs) that facilitate connections with other molecules (glycoproteins or glycolipids) on the surface of cells or in the extracellular matrix[31]. Galectins, however, the most widely expressed lectin subfamily, are freely expressed in the extracellular milieu without membrane-anchoring or glycosylation[32]. In 14 out of the 15 galectins, multivalency is the results of tandem-repeat CRDs or dimer formation. In contrast, galectin-3 is monomeric in solution and has just one CRD, which is tethered to an intrinsically disordered NTD, making it the only chimeric protein in the family. One of galectin-3's functions is to agglutinate organelles, such as neutrophils and laminin[33,34], or glycosylated molecules[35,36]. This function, in which multivalent galectin-3 acts as a "bridge", is lost when the NTD is removed[33,34,37,38]. The mechanism through which galectin-3 becomes multivalent is a matter of debate[37,39–46]. A pentamer model[39] with cross-linking through the NTD has been proposed and is the most widely used[35,36] explanation; however, there are no structural data to support this model and a large body of evidence indicates that galectin-3 is monomeric[42,47–49]. We have previously demonstrated that the intrinsically disordered NTD interacts fuzzily, intermolecularly and intramolecularly, with the region of the CRD not involved in carbohydrate recognition, and that the NTD also self-associates (Fig. 1a–e)[50]. An accumulation of weak self-associations may promote the formation of higher-order oligomers[51]; however, how these weak interactions of galectin-3 contribute to stronger interactions such as cell adhesion and signal transduction is unclear.

In this study, we demonstrate that galectin-3 LLPS is driven by interactions between aromatic residues in the NTD. Using lipopolysaccharide (LPS) micelles as a model template, we show that this mechanism is also involved in full-length galectin-3's extracellular agglutination function. The CRD binds the saccharide moiety on the surface of the micelles, increasing the local protein concentration. The resulting accumulation of multivalent NTD interactions transiently connects the monomers to each other in a manner resembling LLPS.

## Results

**Galectin-3 undergoes LLPS.** To investigate galectin-3's ability to agglutinate, we used LPS micelle as an in vitro assay[40,52]. As shown previously, the higher the protein or LPS concentrations are, the more turbid the samples are (Fig. 1h, i, Supplementary Table 1, Supplementary Table 2)[40,52]. As also reported previously[33,34,37–39,42,47], galectin-3 does not agglutinate in the absence of the NTD (Fig. 1j). This therefore raises the question of how the NTD makes monomeric galectin-3 multivalent (Fig. 1k).

The NTD of galectin-3 is prion-like according to the PLAAC algorithm (Fig. 1f)[53], and has a high probability of forming intermolecular interactions through aromatic residues according to predicted π-π interactions (Fig. 1g)[54]. These properties are common in LLPS proteins[55]. In a previous study, we observed that the NTD of galectin-3 self-associates but the CRD alone does not[50]. Indeed, a 1 mM NTD sample with 150 mM NaCl condenses at higher temperatures and dissolves at lower ones in a reversible manner (Fig. 2a and Supplementary Movie 1), showing lower critical solution temperature (LCST) behavior[56,57]. The coexistence curve could be determined using a temperature ramp search (Fig. 2a). Reducing the protein concentration increased the temperature of the transition boundary, as expected for LCST behavior (Supplementary Fig. 1a). We observed many instances of the condensates fusing (Fig. 2b and Supplementary Movie 2) and fluorescence recovery after photobleaching (FRAP) data also show that they have liquid-like properties (Fig. 2c, Supplementary Fig. 2). Increasing the salt or protein concentration promoted demixing of the protein (Fig. 2d). In the phase-diagram (Fig. 2e), the heterogeneous conditions (indicated as solid blocks) were all confirmed by distinguishing LCST behavior from aggregation. At the highest protein concentration (>2 mM), the NTD sample was homogenous at 15 °C and was heterogeneous at 30 °C in the absence of NaCl (Supplementary Fig. 1b).

We also tested whether full-length galectin-3 undergoes LLPS. We could not exceed concentrations of 600 μM (~16 mg ml$^{-1}$) protein without precipitation and the highest concentration investigated was therefore 500 μM. This sample only demixed at 30 °C when the NaCl concentration was increased above 600 mM. The phase diagram obtained is shown in Fig. 2f. All samples returned to a homogeneous state at 15 °C. Full-length galectin-3 also shows LCST phase separation (Fig. 2g and Supplementary Movie 3) and has liquid-like properties as confirmed by fluorescence microscopy and FRAP experiments (Fig. 2h). We determined that condensation was not the result of salting-out because the CRD alone and a full-length construct without aromatic residues in the NTD remained in a single phase under the same conditions (see below). Furthermore, at a high enough concentration, the NTD phase separated in the absence of NaCl (Supplementary Fig. 1b). The fact that the NTD and full-length galectin-3 undergo LLPS suggests that the multivalent connections involving the NTD also promote agglutination. We therefore investigated the mechanism driving these multivalent interactions.

**Aromatic residues contribute to LLPS of the NTD of galectin-3.** Sequence alignment indicates that the CRDs of galectin-3 are highly conserved among vertebrates (Fig. 3a, orange blocks), and

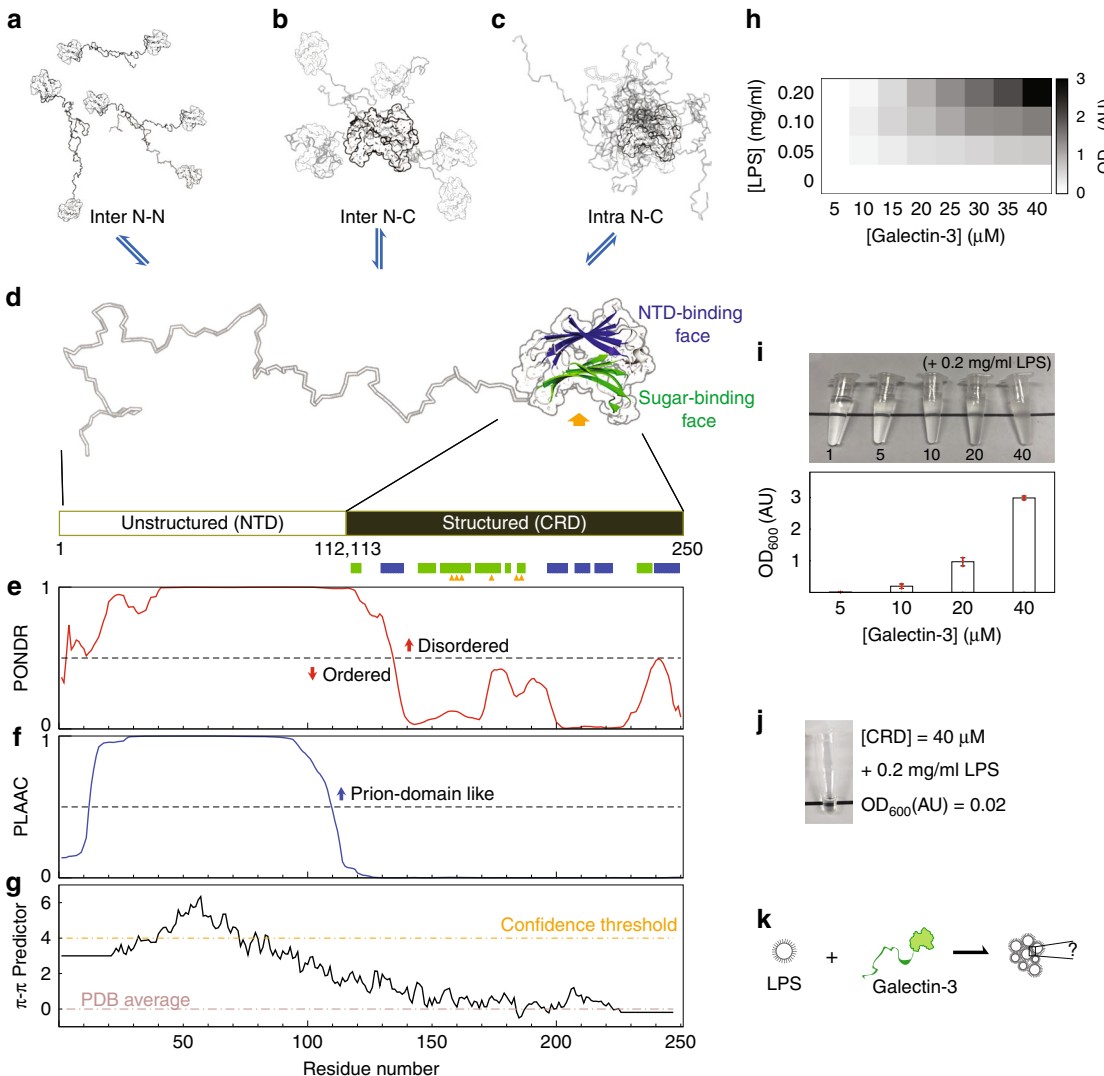

**Fig. 1 Galectin-3 structure, sequence analysis and lipopolysaccharide (LPS) assay. a, b** Intermolecular and (**c**) intramolecular galectin-3 interactions identified in a previous study. **d** A galectin-3 monomer: the structured carbohydrate recognition domain is represented by the solved crystal structure (PDB code, 2NMO), with the (non-canonical) N-terminal domain (NTD)-interaction face (with five β-strands) colored blue and the (canonical) sugar binding face (with six β-strands) colored green. Sugar-specific binding sites are indicated with yellow arrows. **e–g** Sequence analyses: (**e**) level of structural disorder (using PONDR[79]), (**f**) similarity to prion-like proteins (using PLAAC)[53], (**g**) propensity to form π-π interactions[54]. **h, i** Turbidity (O.D$_{600nm}$) of LPS–galectin-3 samples measured at different protein and LPS concentrations. **j** Photograph of the CRD-only sample under the maximum turbidity conditions in panel **i**. **k** A schematic illustration of the main question of this study: what drives monomeric galectin-3 to agglutinate?.

even more highly among mammals (Supplementary Fig. 3). Although the sequences of the intrinsically disordered regions are not similar, two patterns are conserved: proline-glycine repeats followed by aromatic residues (PGxY or PGxW motifs) and a net negative charge. The fact that these patterns are conserved suggests that they are functionally important. The abundance of Tyr or Trp residues has been shown in RNA binding proteins to promote LLPS, such as tyrosines in hnRNP A2 and FUS[58] and tryptophans in TDP-43 (ref. [59]). On this basis, we characterized the self-association tendency of three constructs in which all tryptophans, all tyrosines, or all tryptophans and tyrosines were replaced with glycines (denoted W/G, Y/G, and WY/G, respectively; Fig. 3b).

We showed in a previous study that intermolecular NTD interactions are negligible at protein concentrations lower than 40 µM, but pronounced at 400 µM[50] (in the single-phase regime). We therefore compared the NMR peak intensities of 40 and 400 µM NTD samples in the absence of salt. The $I_{40}/I_{400}$ peak

intensity ratios (Fig. 3c) are higher than the corresponding molar ratio (0.1), indicating that the signals in the spectra from the 400 µM are broader, presumably because of self-association. The average ratio increases with temperature, suggesting that this self-association is driven by hydrophobicity[60]. We did not apply the NMR studies on the sample in the heterogeneous state because the size of the condensate was too large for NMR detection and thus we cannot differentiate that the signal loss is because of forming higher-order oligomers or self-association. Furthermore, high salt concentration also deteriorates the quality of NMR spectrum. For the W/G and Y/G constructs, the intensity ratios are closer to the molar ratio and when all tryptophan and tyrosine residues are removed (WY/G), the intensity ratios are the same as the molar ratio and independent of temperature (Fig. 3c). In agreement with the intensity ratio results, the transverse relaxation rate constants ($R_2$), which reflect the overall effects of tumbling, the internal motions of the molecule, and chemical exchange due to micro-to-millisecond

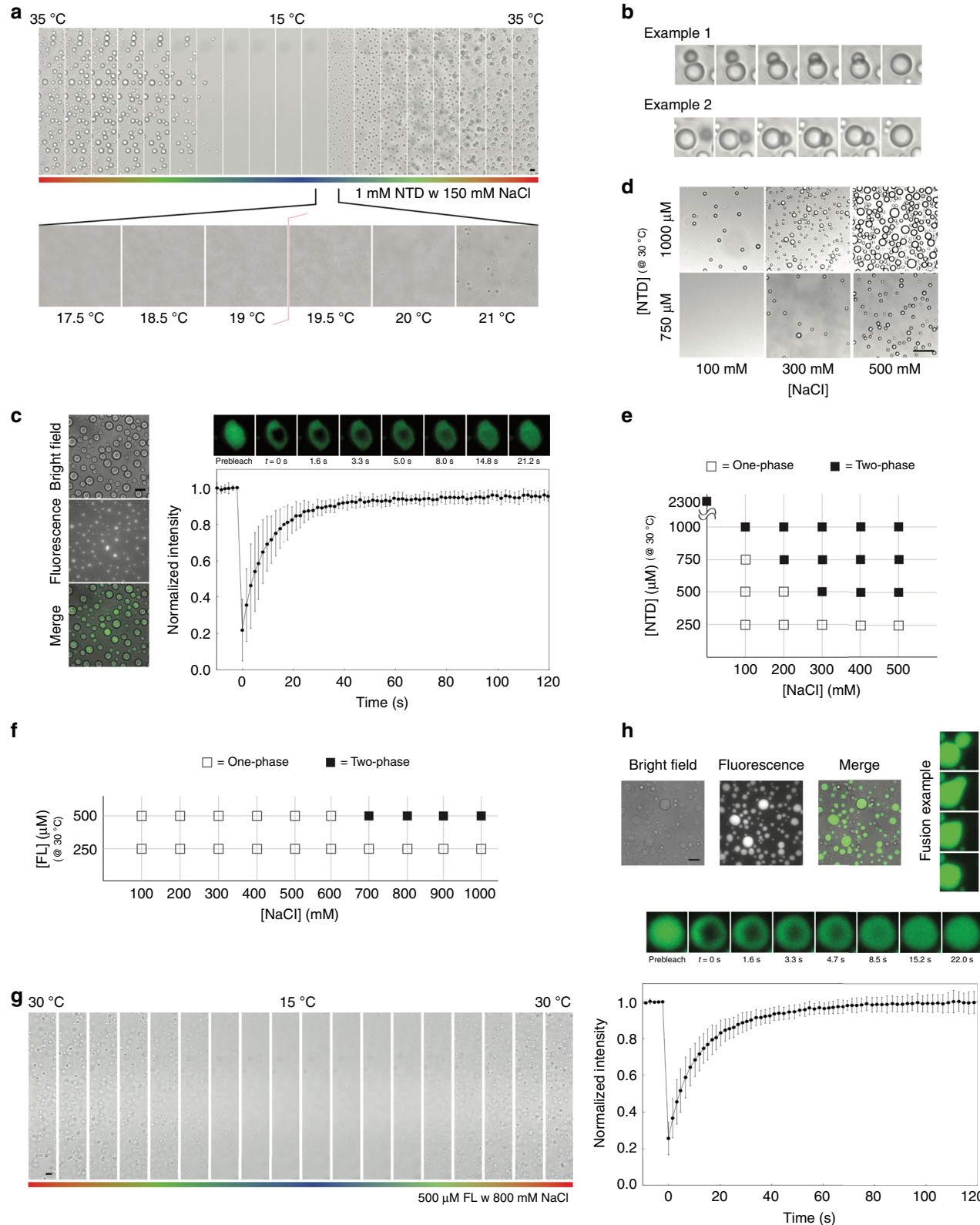

timescale motion[61,62], are concentration dependent for the wild-type NTD, indicating intermolecular association at the higher concentration, but not for the WY/G construct (Fig. 3d). (As demonstrated previously[50], the longitudinal relaxation rate constants, $R_1$, do not vary with the concentration.) For the other constructs, the overall reduction in $R_2$ (cf. the black lines in Fig. 3d and Supplementary Fig. 4c) indicates increased dynamics

and flexibility (Supplementary Fig. 4b). We mapped the NMR chemical shift assignments of the wild-type and the W/G construct based on a previous publication[63]. We did not assign the chemical shifts of the Y/G and WY/G constructs (marked as unassigned in Fig. 3) because only the overall trend in the peak intensity ratios and $R_2$ changes are critical to our interpretation. Furthermore, severe overlap in the NMR spectra of these two

**Fig. 2 Micrographs illustrating the liquid-liquid phase separation (LLPS) properties. a–e** the N-terminal domain (NTD); (**f–h**) full-length galectin-3. **a** A 1 mM sample with 150 mM NaCl condensing reversibly as a function of temperature (35–15–35 °C; scale bar: 10 μm), **b** Examples of condensate fusion. Experiments were performed at least three times for each protein sample. **c** Bright field and fluorescence micrographs alongside the results of fluorescence recovery after photobleaching (FRAP) experiments (scale bar: 50 μm; FRAP recovery cruve is averaged from 10 different condensates; data are presented as mean values +/− SD). **d** Examples of the effects on condensate formation of the protein and salt concentrations (scale bar: 50 μm) and (**e**) the corresponding phase diagram (at a fixed temperature of 30 °C). Experiments were performed at least three times for each protein sample. **f** Phase diagram of full-length galectin-3 as the function of protein and salt concentration. **g** Micrographs showing a 500 μM full-length galectin-3 sample with 800 mM NaCl condensing reversibly as a function of temperature (30–15–30 °C; scale bar: 10 μm). Experiments were performed at least three times for each protein sample. **h** Bright field and fluorescence micrographs alongside the results of FRAP experiments with an example of condensate fusion (scale bar: 50 μm; FRAP recovery curve is averaged from 10 different condensates; data are presented as mean values +/− SD).

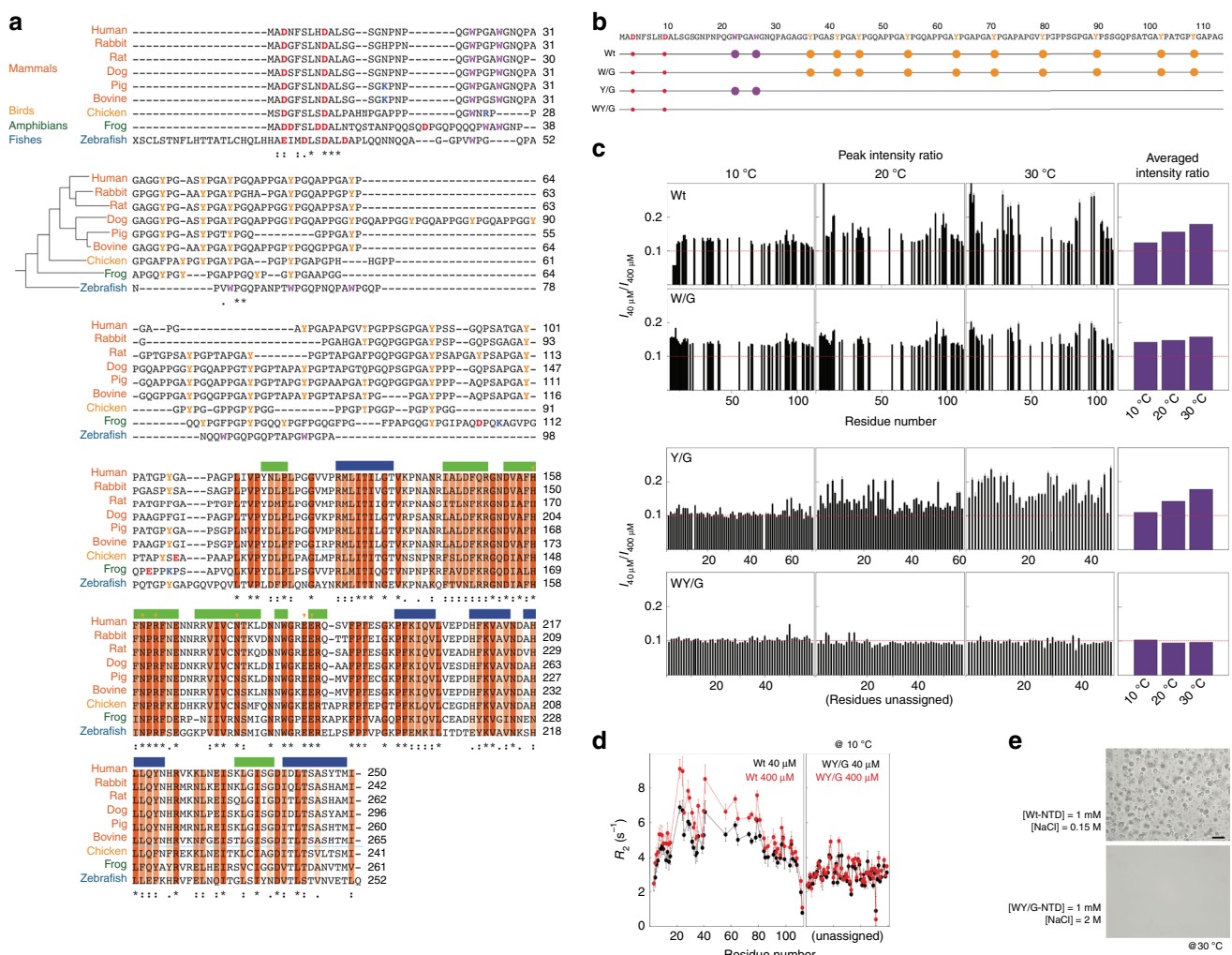

**Fig. 3 The key residues that mediate intermolecular N-terminal domain (NTD) interactions. a** Alignment of galectin-3 amino-acid sequences from different vertebrates (human, five mammals, one bird, one amphibian, and one fish). The phylogenetic tree is plotted based on a summary of multigene and multiprotein studies[87]. Residues in the carbohydrate recognition domain (CRD) are highlighted with dark, medium, or light orange depending on their level of conservation. Negatively and positively charged residues, tyrosines, and tryptophans in the NTD are colored in red, blue, yellow and purple, respectively. **b** Amino-acid sequences of the constructs used in this study with negatively charged residues, tryptophans, and tyrosines highlighted by solid red, purple, and yellow circles, respectively. **c** NMR peak intensity ratios between 40 and 400 μM samples at 10, 20, and 30 °C for the four NTD constructs. The average peak intensity ratios at each temperature are shown in the right-most panel. Red dashed lines indicate the intensity ratio expected from the molar ratio (0.1). **d** Transverse relaxation rate constants for the 40 (black) and 400 (red) μM wild-type and WY/G mutants at 10 °C. **e** Micrographs of samples of wild-type galectin-3 and the WY/G construct under conditions corresponding to the two-phase regime in the wild-type phase diagram (Fig. 2e; scale bar: 50 μm). Experiments were performed at least three times for each protein sample.

constructs (because of the many repeated amino-acid motifs) would have made assignment difficult. Overlapping peaks also provide limited residue-specific information. In these constructs, we therefore only analyzed the well-resolved peaks to avoid bias due to peak overlap.

The reduction in intermolecular self-association indeed abolishes phase separation. At a concentration of 1 mM, the wild-type NTD enters the two-phase regime at salt concentrations above 0.1 M at 30 °C (Fig. 2e) whereas condensates were only observed for the W/G mutant when the NaCl concentration was

2 M (Supplementary Fig. 5). However, even at such a high NaCl concentration, the WY/G construct remains single phase (Fig. 3e). These results indicate that NTD monomers assemble through intermolecular interactions involving aromatic residues.

**The importance of aromatic residues in full-length galectin-3**. The intermolecular and intramolecular interactions between the NTD and the CRD can be distinguished using NMR peaks from the non-carbohydrate-binding site (colored blue in Fig. 1d; NMR spectra in Fig. 4d): The chemical shift differences between the CRD-only and full-length constructs indicate that the NTD and CRD interact intramolecularly via the non-carbohydrate-binding site (blue arrows in Fig. 4d; schematic representation on the right). The concentration-dependent chemical shift perturbations in the non-carbohydrate-binding site for wild-type galectin-3 indicate that the NTD and CRD also interact intermolecularly (orange arrows in Fig. 4d; see ref. [50] for more details). Compared with the wild type, intermolecular and intramolecular NTD-CRD interactions are weaker in the W/G mutant (Fig. 4a; comparison with wild-type spectra in Supplementary Fig. 6a) and more obviously so in the Y/G mutant (Fig. 4b), indicating that the two remaining tryptophans and the ten remaining tyrosines contribute to self-association. The relative contributions of the two types of aromatic residues cannot be evaluated because their numbers differ. The NMR peaks from the 40 and 400 μM full-length WY/G mutant all overlap, indicating that there is no self-association (Fig. 4c, e). Note also that the peaks from the CRD in the full-length WY/G construct overlap with the corresponding peaks in the CRD-only spectrum (Fig. 4c), indicating that the NTD and the CRD do not interact. The $R_2$s of the full-length WY/G construct are likewise similar at the two concentrations, in contrast with the wild type (Fig. 4f, h), leading to the same conclusion. Furthermore, the average $R_2$ is lower when the aromatic residues are removed, indicating increased motion in the protein, presumably because there is less chemical exchange driven by intermolecular and intramolecular interactions (Fig. 4f, h and Supplementary Fig. 6b, c). Although the aromatic residues do not contribute to the constructs' level of disorder and prion-likeness (Supplementary Fig. 7), the loss of them reduces their predicted propensity to form π–π interactions (Fig. 4i). Removal of the aromatic residues also prevents LLPS in full-length samples. No condensates were observed for the 500 μM WY/G construct with 1 M NaCl at 30 °C (Fig. 4j), conditions under which the wild-type sample is in the two-phase regime (Fig. 2f).

**Multivalent connections in galectin-3/micelle agglutination**. Samples of full-length galectin-3 and of the NTD alone become heterogeneous at non-physiological protein or NaCl concentrations (Fig. 2). However, the function studied here is galectin-3's ability to agglutinate. Galectin-3 binding on the surface of LPS micelles increases the local protein concentration to levels in the tens of mM range (see estimate in the Supplementary Note 1), much higher than in the samples prepared here. The fact that the NTD and the full-length construct undergo LLPS suggests that multivalent crosslinking contributes to galectin-3's agglutination function. Just as we did for full-length galectin-3 (Fig. 1i), we used LPS micelle assays to investigate agglutination in the galectin-3 constructs: the full-length W/G mutant also became turbid but to a lesser extent, but the CRD-only, NTD-only, full-length Y/G and WY/G samples remained transparent when mixed with LPS (Fig. 5a, b; Supplementary Table 3; adding NaCl had little effect on the turbidity of the samples, Supplementary Fig. 8a).

The NMR peak intensities with and without LPS for the full-length wild type and the NTD-only, CRD-only, and full-length WY/G constructs (40 μM samples with 0.2 mg/ml LPS) are compared in Supplementary Fig. 8b. For the full-length wild type, the signal intensity in the presence of LPS was too weak for further analysis (Supplementary Fig. 8b, the only signals detected were from the flexible NTD domain) because the large size of the protein–micelle aggregates leads to severe line broadening. Although the NTD-only, CRD-only, and WY/G constructs remained transparent when mixed with LPS (Fig. 5a), different intensity ratio distributions were measured for the three samples (Fig. 5c). For the NTD-only construct, the fact that the peak intensity ratios are all about 1 indicates that there is no interaction with the micelles (Fig. 5c, d). The significantly lower peak intensities in the presence of LPS for the CRD-only construct (Fig. 5c) can be explained by a substantial proportion of the molecules binding to the polysaccharides of LPS and thus those CRDs' NMR signal is undetectable because of the reduced overall tumbling rate of the protein on the large LPS micelles (Fig. 5d). The same mechanism explains why the peaks from the CRD in the WY/G mutant decrease in intensity in the presence of LPS (Fig. 5c), but those from the NTD do not, as the CRD interacts with LPS while the NTD remains highly dynamic (Fig. 5d). Figure 5e shows the model proposed on the basis of these results for the interaction between full-length wild-type galectin-3 and LPS. When LPS micelles loaded with galectin-3 come into contact, the NTDs interact through their aromatic residues (the mechanism that drives LLPS in the absence of LPS; Fig. 2), leading to agglutination, as depicted in Fig. 5e. The turbidity of the LPS/galectin-3 mixtures is proportional to the protein concentration (Fig. 1h), which is consistent with this velcro-like behavior of the NTD: the more galectin-3 molecules are attached to each micelle, the more they tend to agglutinate.

Adding 25 mM lactose, a commonly used ligand for galectin-3, dissolves the galectin-3–LPS mixture (40 μM and 0.2 mg ml$^{-1}$, respectively; Fig. 6a). The NMR spectrum of this sample has peaks of the same intensity and at the same chemical shifts as that of the same concentration of galectin-3 and lactose without LPS (Fig. 6b, c), indicating that galectin-3 returns to the monomeric state even after having agglutinated LPS. The chemical shift changes in the residues around the carbohydrate-binding face confirm the binding of lactose (Fig. 6d). Our results indicate that the NTD remains highly dynamic at all stages of the process and that the interactions are transient (Fig. 6e). When the CRD is bound to LPS, this increases the local protein concentration and the transient interactions between the NTDs promote agglutination. When the interaction between the CRD and LPS is blocked by lactose, the local concentration is too low to maintain the interactions between the NTDs and the protein molecules return to their monomeric state (Fig. 6e). We cannot rule out the possibility that interactions between lactose and the CRD also cause a conformational change that reduces the propensity to phase separate as there were chemical shift perturbations in the non-carbohydrate-binding site (Fig. 6d). These lactose experiments indicate that unlike the other members of the galectin family, which form dimers or are tandem repeats, free galectin-3 is monomeric in solution, without preformed oligomers.

## Discussion
The functional repertoire of biomolecular condensates has only recently began to come to light[55]. This repertoire is not limited to intracellular functions however. LLPS is thought to initiate the assembly of many proteins in the extracellular matrix[64], as studied using model peptides designed to mimic elastins[65]. The intrinsically disordered NTD of galectin-3 has many repeated proline-glycine motifs, which are also found in elastin-like proteins. However, the prevalence of aromatic residues in the NTD is not typical of elastins but of proteins known to undergo LLPS.

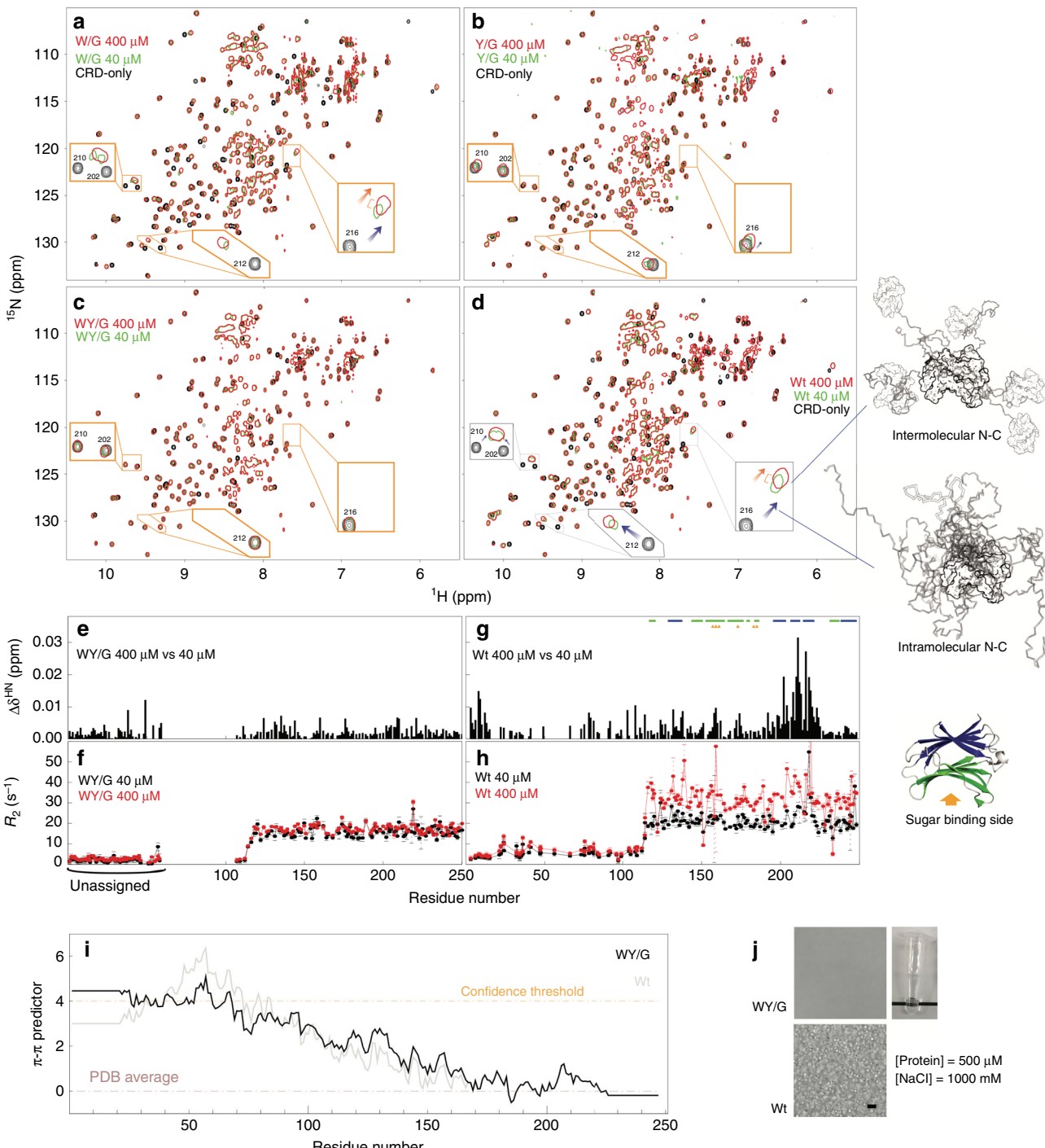

**Fig. 4 Intermolecular and intramolecular interactions and liquid-liquid phase separation in full-length galectin-3. a–d** NMR spectra of 400 (red) and 40 (green) µM samples of the four full-length constructs of galectin-3 compared to those from the CRD-only construct (black, 400 µM). Expanded views are shown of the regions in which the peaks shift the most. Blue arrows indicate chemical shift differences due to intramolecular NTD-CRD interactions (full-length spectra vs. the CRD-only spectrum); orange arrows indicate chemical shift differences due to intermolecular NTD-CRD interactions (400 vs. 40 µM spectra). Schematic illustrations of these interactions are shown on the right of the figure. **e, g** Chemical shift and (**f, h**) transverse relaxation rate constant differences between 400 and 40 µM samples for the WY/G mutant and the wild type. **i** Propensity to form π–π interactions of the WY/G construct. **j** Photograph and micrographs (scale bar: 10 µm) of samples of the full-length galectin-3 variants under conditions corresponding to the two-phase regime of the phase diagram of full-length wild-type galectin-3 (Fig. 2f). Experiments were performed at least three times for each protein sample.

Furthermore, while positively charged lysines (responsible for covalent crosslinking) are prevalent in elastin-like proteins, there are few charged residues in galectin-3's NTD, with a net negative charge in all vertebrates (Fig. 3a), suggesting that the NTD, although disordered, serves a function. Our data (Fig. 2) confirm the propensity to assembly suggested by prion-likeness and π–π interaction sequence analysis, with the aromatic residues acting as "stickers"[6,66] (Fig. 3). These weak contacts lead to the formation of stronger assemblies when the protein concentration (i.e the number of stickers) is above the phase transition boundary. Our

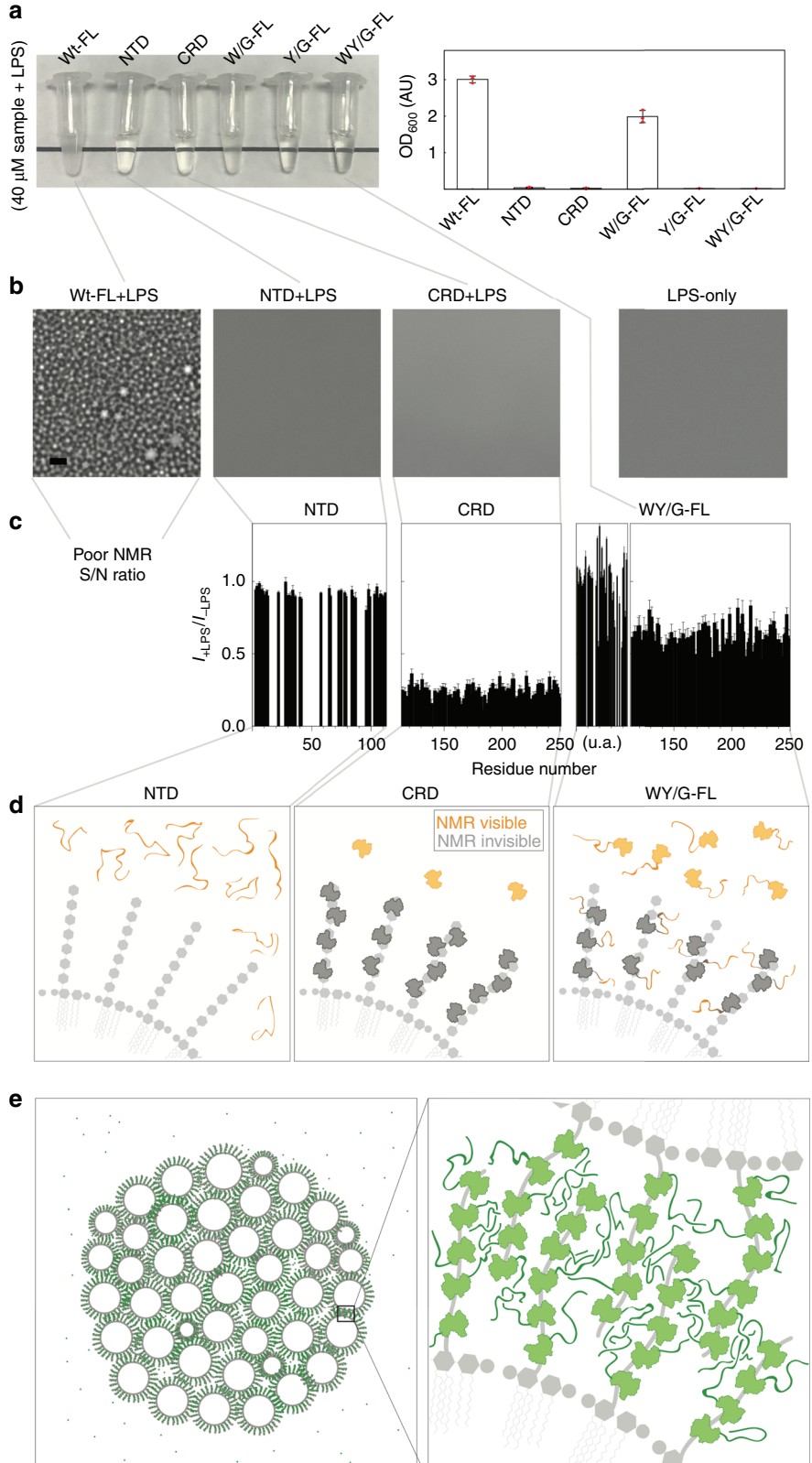

**Fig. 5 Lipopolysaccharide (LPS) micelle agglutination assays. a** Photograph and turbidity (O.D$_{600}$) ($n = 3$ independent experiments; data are presented as mean values $+/-$ SD) and (**b**) micrographs of samples of the different galectin-3 constructs in the presence of LPS. Experiments were performed at least three times for each protein sample (Scale bar: 10 μm). **c** NMR peak intensity ratios in the presence and absence of LPS plotted as a function of residue number for the N-terminal domain (NTD)-only, carbohydrate-recognition domain (CRD)-only, and WY/G constructs. (This analysis was not possible for the full-length wild type because of a poor signal-to-noise ratio (Supplementary Fig. 8). **d** Schematic illustrations explaining the intensity changes for the NTD-only, CRD-only, and WY/G constructs. **e** Schematic illustration of the proposed mechanism of LPS-micelle agglutination driven by full-length wild-type galectin-3.

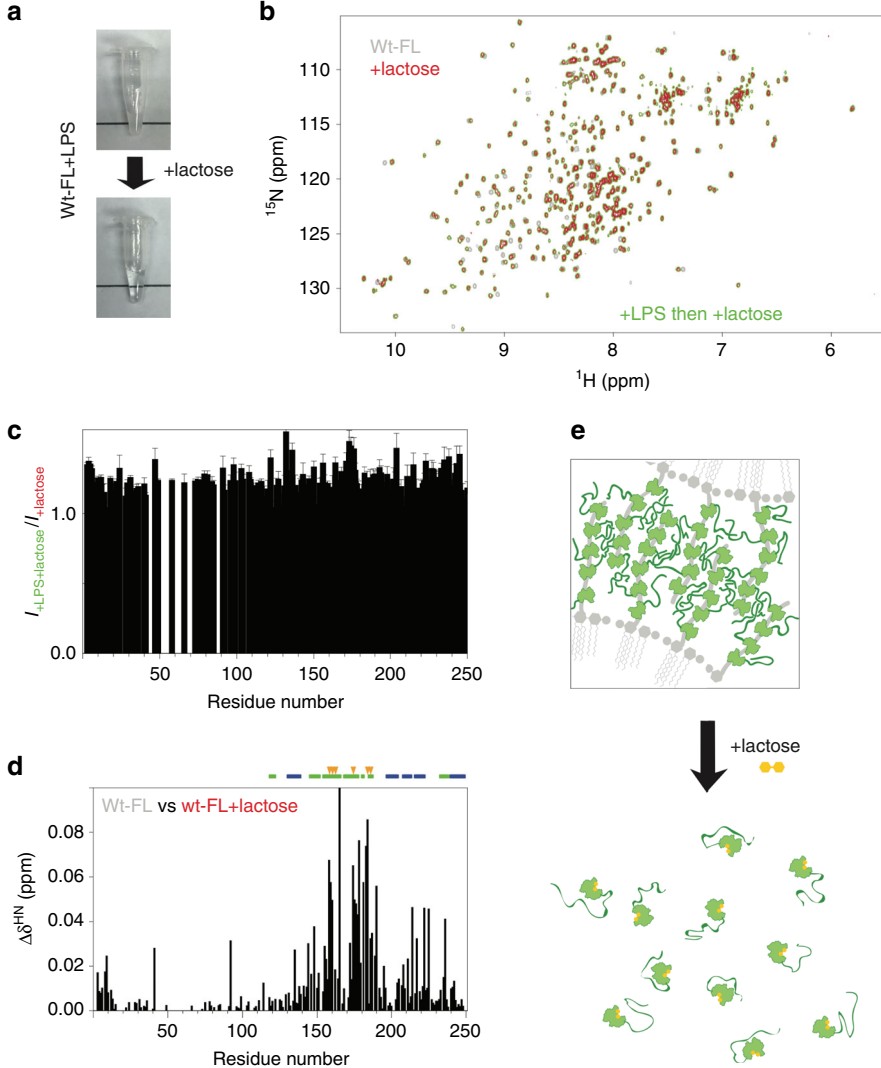

**Fig. 6 Interaction with lactose inhibits galectin-3 agglutination. a** Photographs of a galectin-3 (40 μM) and LPS micelles (~0.2 mg ml⁻¹) sample before and after adding 25 mM lactose, which disrupts the aggregates. **b** Comparison of the spectra of 40 μM samples of full-length wild type without LPS before (gray) and after adding 25 mM lactose (red), or mixed with LPS in the presence of 25 mM lactose (green). **c** NMR peak intensities of the LPS/galectin-3 sample recovered with 25 mM lactose normalized to the corresponding peak intensities from a sample of galectin-3 with the same amount of lactose but no LPS. **d** Chemical shift differences between the spectra recorded in the presence vs. in the absence of lactose (red (or green) vs. gray in panel **b**). The most pronounced changes in chemical shift occur for residues in the carbohydrate binding site (indicated with orange arrows) confirming the binding of lactose. **e** Illustration of how lactose dissolves the galectin-3/LPS aggregates.

analysis also shows that the aromatic residues also contribute to LLPS, intramolecular interactions, and (micelle) agglutination in full-length galectin-3 (Figs. 4, 5). In the model we propose (Figs. 5, 6), the CRD binds to surface glycans, exposing the NTD and promoting intermolecular interactions, which lead to the formation of micelle aggregates when the local concentration of NTDs is high enough to maintain strong connections. However, because the interactions are transient and dynamic, the protein-micelle aggregates return to the monomeric state when connections between the two are prevented (Fig. 6e). These results add an extracellular process to the functional repertoire[55] of LLPS.

LLPS can be artificially induced in some cases to demonstrate how the propensity to aggregate is related to function or disease, as shown for example in eye lens crystallins[67,68] and in disease-related hemoglobin mutants[69]. Although galectin-3 only undergoes LLPS under non-physiological conditions, it agglutinates in the extracellular matrix through interactions with cell surface ligands that locally increase its concentration. This agglutination may involve the same multivalent mechanism as the one that drives LLPS and is reminiscent of the intrinsically disordered regions of nucleoporins (the FG-repeats), which are locally concentrated in the passage of the nuclear pore complex. This FG-repeat region is also regarded as a biomolecular condensate[2] and their phase properties have been studied[70]. A similar effect is described in a recent study of the m⁶A-binding protein YTHDF[71]: YTHDF only functions when mRNA is poly-methylated such that many copies of YTHDF are recruited and the local concentration becomes high enough for the IDR to undergo LLPS. With singly methylated mRNA on the other hand, the single weakly bound YTHDF molecule does not undergo LLPS and is not functional. Our model is also consistent with known behaviors of galectin-3: in agreement with our turbidity vs. concentration results for example (Fig. 1i), galectin-3 adhesion increases with concentration in many types of inflammatory

cells[33]. How an accumulation of weak interactions can lead to stronger cell-cell adhesion via the NTD is illustrated in Supplementary Fig. 9. Although many studies have shown that removing the NTD leads to loss of function[33,37,47], our model shows more specifically how LLPS is initiated by interactions between aromatic residues in the NTD. Galectin-3, unlike dimeric and tandem repeat galectins, may have evolved multivalency through an LLPS-like mechanism, with the potential advantage that agglutination depends on the amount of protein present in the extracellular milieu.

Fuzzy interactions also facilitate crosslinking between galectins and glycolipids or glycoproteins to form the glycan-galectin lattice, a component of many mechanisms such as the regulation of receptor kinases and inflammatory response[72]. Inserting our model into these molecular processes yields the following scenario: galectin-3 molecules attached to glycoproteins on membrane rafts come into contact by diffusion. When the galectin-3 concentration is low, the weak NTD-NTD interactions do not stabilize the glycan-galectin lattice; however, if enough galectin-3 molecules are attached, galectin-3 agglutinates the membrane glycoproteins, inducting[73] or inhibiting[74] downstream signaling (Supplementary Fig. 9b, c). Furthermore, the mechanism described here of how galectin-3 agglutinates is a potential target for drug design. Instead of designing compounds to selectively inhibit galectin-3's carbohydrate binding site (see ref. [75] for example) rather than the CRDs of other members in the galectin family, targeting the aromatic residues in the NTD may also be a good strategy.

Our current study focuses on interpreting the extracellular function of galectin-3 but it seems probable that the NTD's ability to phase separate may also contribute to many of its intracellular functions, such as sensing danger by detecting unusual glycan exposure on damaged phagosomes[76] or by regulating RNA splicing[77] as do many RNA binding proteins with a prion-like domain (e.g., FUS, TDP-43). Moreover, many of the mechanisms governing the behavior of galectin-3 are still puzzling. For instance, how is the equilibrium between the N and C terminal domains (Fig. 1c vs. d) mediated? Does the level of negative charge play a role in the mediation of LLPS by phosphorylation (which adds more negative charge; phosphorylation sites have been identified[78])? Do galectin-3's interactions with binding partners through the non-canonical carbohydrate-binding site outcompete the NTD-CRD interaction and thereby release the NTD to regulate LLPS? It is evolutionarily unlikely that galectin-3's disordered tail is redundant and this article demonstrates how its multivalent interactions mediate function. Many mysteries about this chimeric protein remain and our study provides a framework for further investigations.

## Methods

**Sequence analysis**. The level of structural disorder was predicted using the PONDR VLXT algorithm[79]. Prion-likeness was predicted using the PLAAC algorithm[53] and $\pi$–$\pi$ interactions using the Pi-Pi prediction server (PScore Predictor)[54]. The protein sequences were aligned using BLAST (www.uniprot.org/blast). The entries used to access the sequences from the database are listed in Supplementary Table 4.

**DNA constructs**. The cDNA of galectin-3 variants (full-length or NTD-only) was inserted into a pHD-vector containing a hexahistidine-tagged small ubiquitin-like modifier protein (His$_6$-SUMO) as described previously[50]. The WY/G construct was created by whole gene synthesis (Supplementary Note 2). The primers used for the W/G (tryptophan changed to glycine on the wild-type template) and the Y/G (tryptophan changed glycine on the WY/G template) construct are listed in Supplementary Table 5. The cDNA of GFP was inserted at the C-terminus of the His$_6$-SUMO-NTD construct. All constructs were verified by DNA sequencing.

**Protein expression and purification**. All variants were purified using the same previously-described protocol[50]. A fusion protein of hexahistine, SUMO, and the

galectin-3 variant (His$_6$-SUMO-gal3v) was purified using a nickel-charged immobilized metal-ion affinity chromatography (IMAC) column (Qiagen, Inc.). The column was washed using 10 column volumes (CVs) of 50 mM Tris-HCl with 300 mM NaCl at pH 7.5, and the bonded-protein was eluted using five CVs of the same buffer with an additional 500 mM imidazole. Imidazole was removed using a PD-10 column (GE Healthcare, Inc.). 6xHis-Ulp1[403–621] protease was added to the protein solution with a final concentration of 30 μM and left at 4 °C for 2 h to detach the 6xHis-SUMO tag and galectin-3. The protease-digested solution was loaded into a nickel-charged IMAC column, and the flow-through was collected. The target protein was concentrated using a Centricon centrifugal filter (Vivaspin) and was exchanged with phosphate buffer (20 mM) at pH 6.8 using a PD-10 column (GE Healthcare). Protease inhibitor (Roche Applied Science) was added before storage. The purified NTD sample was flash frozen with liquid nitrogen and stored at −80 °C until needed. The full-length samples were stored at 4 °C for 2 days at most before experiments.

**Microscopy**. Micrographs were collected using an Olympus BX51 device equipped with a ×40 long working distance objective. The images were recorded using a Zeiss AxioCam MRm camera. The protein samples were placed on a thermostatic stage (THM120, Linkam Scientific Inc.). The sample was equilibrated in this chamber for at least 3 min before each measurement.

**Fluorescence recovery after photobleaching (FRAP) assays**. Fluorescence recovery after photobleaching experiments on GFP-tagged proteins were performed using a 100× 1.49NA Plan objective lens attached to an iLas multimodal total internal reflection fluorescence (Roper)/spinning disk confocal (CSUX1, Yokogawa) microscope (Ti-E, Nikon). The stage temperature was maintained at 37 °C with an airstream incubator (Nevtek) and focus was maintained using the PerfectFocus[(TM)] system (Nikon). The 488-nm laser was used to photobleach the fluorophores into a single fluorescent droplet. Images were acquired at 1 s intervals before and after photobleaching using an Evolve EMCCD camera (Photometrics) with an ~100 nm evanescent field depth. Images were captured and processed using the Metamorph software. Samples of NTD or full-length galectin-3 at 500 μM in 500 mM or 1 M NaCl were mixed with 1% (5 μM) of NTD-GFP or with 1% GFP as control.

**NMR experiments and analysis**. NMR data were collected on Bruker AVIII 850-MHz or 600-MHz spectrometers, both equipped with a TCI cryogenic probe. The $^1$H-$^{15}$N HSQC spectra and transverse relaxation rate experiments were collected using standard pulse sequences[80–82]. In the dynamics experiments, the transverse relaxation rates were measured with delays of 17, 34, 51, 68 ms for the first four time points and 85 and 102 ms (for the full-length constructs) or 119 and 153 ms (for the NTD-only variants) for the last two. Peak intensities were fitted to exponential decays with a Monte Carlo procedure to estimate fitting error. The dynamics data were collected in an interleaved manner with an interscan delay of 3 s. All NMR data were collected at 30 °C, unless otherwise stated. The samples were prepared in 20 mM phosphate buffer at pH 6.8 containing protease inhibitor (Roche Applied Science) and 10% D$_2$O.

All data were processed using NMRPipe[83] and analyzed with SPARKY[84]. Peak intensities and errors were determined using the non-linear line-shape analysis (nlinLS) function in NMRPipe based on the noise of the spectra. The intensity ratios were normalized to the number of scans and the errors were calculated using standard error propagation. The average chemical shift difference ($\Delta\delta^{av}$) was calculated using

$$\Delta\delta^{av} = \sqrt{\frac{(\Delta\delta_H)^2 + (\frac{1}{5}\Delta\delta_N)^2}{2}} \quad (1)$$

where $\Delta\delta_H$ and $\Delta\delta_N$ are the differences in chemical shift between two $^1$H-$^{15}$N HSQC spectra for the amide proton and nitrogen, respectively.

**LPS assay**. Lipopolysaccharide from E. Coli strain O55:B5 was purchased from Merck (Cat. L2880). A stock solution was prepared in 20 mM phosphate buffer at pH 6.8. The chosen amounts of LPS to be above the critical micelle concentration[85,86] and protein solution were mixed directly before optical density or microscopy measurements. All measurements were performed in triplicate. For the NMR experiments comparing the behavior of galectin-3 with and without LPS, a batch of stock protein solution (at high concentration) was separated into two parts; LPS was added to one and the same volume of buffer to the other.

**Reporting summary**. Further information on experimental design is available in the Nature Research Reporting Summary linked to this paper.

## Data availability

The source data underlying Figs. 1h, 1i, 2c, 2h, 3c, 3d, 4e–h, 5a, 5c, 6c, and 6d and Supplementary Figs. 4c, 6c, and 8a are provided as a Source Data file. Other data are available from the corresponding author upon reasonable request.

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

## Acknowledgements
The authors thank Prof. Won-Jing Wang (NYMU) for access to the microscope, Prof. Fu-Tong Liu (Academia Sinica) for helpful comments, and the Core Facility for Protein Structural Analysis in Academia Sinica for access to the NMR spectrometers. This work was supported by the Ministry of Science and Technology of Taiwan (106-2113-M-010-005-MY2 and 108-2113-M-010-005).

## Author contributions
J.R.H. conceived the project and wrote the paper. Y.P.C., Y.C.S., D.C.Q., Y.H.L., and J.R.H. designed the experiments. Y.P.C., Y.C.S., D.C.Q., Y.H.L., Y.Q.C., J.C.K., and J.R.H. collected and analyzed the data.

## Competing interests
The authors declare no competing interests.
