## [Peer Review File · Nature Communications]

Reviewers' comments:

Reviewer #1 (Remarks to the Author):

In this study Chiu and colleagues investigate the oligomerization and liquid-phase separation of galectin-3 and its relation to galectin-3 agglutination. Galectin-3 is in this respect particularly interesting because it contains only a single carbohydrate-recognition domain, while other galectins contain tandem CRDs that mediate agglutination. Instead, galectin-3 contains an intrinsically disordered N-terminal domain, which has a sequence with prion-like properties and thus might undergo liquid-liquid phase separation. Overall this is a very interesting study that is summarized in a well-written and nicely illustrated manuscript.

Major points:

- LLPS is assessed by contrast microscopy and turbidity measurements. However, these are prone to artefacts (e.g. formation of water droplets, non-LLPS oligomerization, ...). Therefore it is important that LLPS of both the NTD and full-length galectin-3 is studied using fluorescent microscopy. In addition, the liquid-like properties of the droplets should be demonstrated by FRAP experiments.
- LLPS data are mostly shown for the NTD and some small experiments are reported later in the manuscript for full-length galectin-3. More data (temperature/ionic strength dependence, sensitivity to aliphatic alcohols, fluorescence microscopy) should be provided for full-length galectin 3 and shown early on in the manuscript together with the data for the NTD.
- Most/all NMR experiments/data reported in the manuscript were recorded under conditions, in which no LLPS occurs. This is a bit weird given that a major focus of the work is on LLPS. Thus, NMR experiments of NTD and full-length galectin-3 in conditions of LLPS should be included, in order to support the conclusions of the manuscript.
- Statements regarding "the first reported example of extracellular LLPS ..." should be removed; please see Kostylev et al. Mol Cell 2018.

Reviewer #2 (Remarks to the Author):

Please see the attached file.

Reviewer #3 (Remarks to the Author):

This manuscript builds on previous observations by the authors of LLPS in galectin-3, a lectin with a disordered, low-complexity N-terminal domain and a C-terminal carbohydrate binding domain. Here, the authors use NMR spectroscopy and microscopy to characterize the influence of salt, temperature and protein concentration on LLPS in this system, as well as to demonstrate the role of aromatic residues in driving LLPS of the N-terminal domain. They also provide some initial data suggesting a role for LLPS in the biological functions of galectin-3, by providing a mechanism for multivalency and agglutination, in the absence of multiple CBDs or an oligomeric protein.

This is in general a well written manuscript that provides new insight, particularly into the mechanism of galectin-3 LLPS. The experiments are well carried out, and the link between the data and the conclusions is clear. It adds to the growing story of lectin LLPS, and also provides a new example of a functional extracellular protein LLPS, for which there are relatively few examples known so far. There are, however two areas in which additional detail or experiments would allow for more concrete conclusions to be drawn, strengthening the manuscript.

1. It is not clear why large sections of some constructs are listed as unassigned. For example, the HSQC data for the Y/G and WY/G NTD constructs look well-resolved, and have favorable R2

values, yet are unassigned. If there are other factors at play, the authors should specify them, and if not, then complete assignment can only improve the data presented. Without sequential resonance assignments, then plotting the R2 or delta-chemical shift data for these constructs as a comparison to assigned data sets is not particularly useful, rather only the aggregate or average values hold meaning. Complete assignments might also provide some additional site-specific information regarding preferred sites for inter/intramolecular interaction.

2. The data shown in Figure 2 suggests the potential to develop a true phase diagram, rather than the cartoon presented in Figure 2E. If feasible, the authors should consider quantifying the monomer vs phase separated fractions as a function of temp/salt/protein, to provide a more quantitative analysis in this panel.

Reviewer #2 (Remarks to Author):

The manuscript by Chiu et al. describes a mechanism by which galectin-3 can undergo liquid-liquid phase separation in extracellular space to mediate agglutination of lipopolysaccharide micelles. They utilize NMR to probe galectin-3 dynamics and inter- and intramolecular interactions between its N-terminal Domain and its Carbohydrate Recognition Domain. Solution turbidity assays are coupled with NMR to discern the amino acid residues that drive galectin-3 LLPS. They state that the galectin-3 CRD serves to increase local protein concentrations in the space between adjacent lipopolysaccharide micelles by binding to micelle carbohydrates and galectin-3's disordered NTD drives phase separation of galectin-3 and subsequent agglutination of LPS micelles. The combination of high-resolution NMR with macroscopic phase separation phenomenon is appreciated and a strong component of the paper, as is the sequence analysis.

The general premise of this paper is highly intriguing, and I would say that I tend to think the author's model may be partially if not entirely correct. However, my major concern in how the study is presented is the concentration regimes used of both protein and NaCl used throughout this study (for different reasons!). I raise this as a concern, not because I believe this discounts the authors' model, but because instead I believe it sets up an incredibly simple 'straw man' that would be used to dismiss the authors' findings – namely that the conditions are so far from physiological relevance that the phenomena studied are simply an artefact of those conditions. At an absolutely minimum, a quantitative protein vs. salt phase diagram (the low-concentration arm of the binodal only) must be provided to allow the reader to understand the salt/protein concentration dependence. The authors hint at this in Figure 2e, but without actual data this sketch is insufficient to draw conclusions from. That said, I do believe an interesting narrative emerges from this work (hence the length of my review!), and I'd like to provide context for my concerns.

Firstly, why are the solution conditions problematic? Essentially every protein will undergo LLPS under the 'appropriate' conditions. Consequently, the observation of LLPS should not be considered evidence of biological relevance. For example, Figure 2a demonstrates the temperature dependence of 1 mM protein solution under semi-physiological salt conditions (150 mM NaCl). Even a small decrease in the concentration to 750 μ M (still 1-2 orders of magnitude higher than the protein concentration required for other systems studied *in vitro*) yields loss of phase separation (Figure 2c) suggesting this is below the (salt-dependent) saturation concentration. Figure 2d further makes the point that an extremely high concentration of protein *and* salt is required for LLPS to be observed. The salt dependence especially is concerning; given the remarkably small number of charged residues, it seems more likely that the high salt concentration is driving a 'salting out' transition, as opposed to a screening of repulsive electrostatics.

The authors then add 10% PEG-8000 (this is a considerably high concentration, given the more standard values are in the 0.5-2% range) which - as may be expected - decreases the saturation concentration into a more physiologically reasonable range. However, I have concerns regarding the molecular mechanism underlying the effect of PEG here. It is worth noting that much work (from Tom Record and others) has demonstrated that PEG cannot be treated as a purely excluded volume crowder, and there is a growing concern about the use of PEG in the context of phase separation studies. I bring this up only to discourage the authors from using 10% PEG-8000 as a tool to "demonstrate" a physiological relevant concentration regime.

However, the authors do then demonstrate that that full-length WT protein + LPS at a protein concentration of 40 μ M protein, a much more reasonable protein concentration, but with (seemingly) no additional NaCl beyond the PBS buffer? However, assuming this result is robust to the absence of NaCl, This for me, and specifically Figure 5a, is the most interesting result – a protein that fundamentally does not undergo LLPS particularly strongly will

robustly do so in the presence of its binding partner in a manner that is dependent on *both* the disordered N-terminal domain and the LPS binding CRD.

With this in mind, I would strongly recommend the authors consider a slightly alternative narrative to the one presented. Explicitly make the point that while monomeric galectin-3 *can* undergo phase separation and that phase separation appears to be dependent on the Trp and Tyr residues, highly non-physiological conditions are required. This allows the authors the opportunity to dissect that sequence-determinants of phase behaviour, while recognizing that the conditions are somewhat ridiculous. In this, a true protein/salt phase diagram would be necessary for this to be a sufficiently rigorous study. Having then established that phase separation *can* occur, but requires these somewhat extreme solution concentrations, the authors can then ask if this behaviour is simply a quirk of the underlying physical chemistry, or might there be functional relevance to this behaviour?

From this the authors can introduce the LPS micelles and propose the model that the micelles provide the requisite higher-order multi-valency to drive phase separation. In effect, the galectin-3 does not (readily) undergo phase separation, instead, micelles coated by galectin-3 (where this coating is dependent on the ability if galectin-3 to bind LPS) are truly the basic units that undergo phase separation. In this model, galectin-3 only undergoes phase separation after forming a supramolecular assembly of LPS micelle coated in galectin-3. The authors should calculate the (theoretical) local concentration of galectin-3 bound to the surface of the LPS micelles and use this to compare with real bacterial to ask if this is (putatively) a relevant biological mechanism. I should say, this is *basically* the same narrative as the authors have now; the key difference in my opinion is explicitly recognizing that in the absence of a supramolecular assembly phase separation of galectin-3 requires entirely non-physiological conditions to occur [BUT this still allows a dissection of the relevant molecular interactions].

In short, the study here does not, in my opinion, provide compelling evidence that galectin-3 (by itself) undergoes phase separation under physiological conditions, which is 100% OK! However, it *does* suggest that if galectin-3 can be locally concentrated (e.g. on the surface of bacteria) this may enable the saturation concentration for phase separation to become sufficiently low. It's not clear to me if agglutination can *actually* be described using phase separation as a physical framework (even just *in vitro*). To argue that galectin-3 drives agglutination through phase separation would require the demonstration that *in vitro* agglutination shows typical hallmarks of phase separation (e.g. a saturation concentration). This may not be technically possible to achieve, in which case the narrative should state that galectin-3 can undergo phase separation under very high protein/salt concentrations, and those same interactions drive agglutination in a manner that is essentially dependent on the recruitment of galectin-3 to the surface of LPS micelles (or to gram-negative bacteria). This is not a problem and does not undermine the study – a functional two-phase system is not the only interesting mode of assembly!

What I don't know is if this model makes any kind of biological sense and it is the author's responsibility to push this further – i.e. in the density of LPS on the surface of gram-negative bacteria sufficiently high to drive assembly in a manner analogous to that observed for LPS micelles? This most explicit test of this would be to ask if the WY/G version is able to drive agglutination of some gram negative bacteria (e.g. *E. coli* - the expectation being no) but I fully recognize this is likely beyond reasonable scope for revisions in this manuscript.

An alternative explanation is that phase separation is an unavoidable consequence of the galectin NTD being sufficiently multivalent to interact with (and modulate?) the behaviour of the CRD. It is difficult to rule this mechanism out (or in) without disrupting the phase behaviour and performing *in vivo* studies (which goes beyond the reasonable scope of this paper) but the possibility of phase separation here being simply an epiphenomenon should be mentioned in the discussion.

Finally, there is language in the paper that makes me wonder if the authors fully understand the underlying physical concepts surrounding a two-phase equilibria (i.e. binodal curves, saturation concentrations, the equalization of chemical potential across the two phases *etc.*). This is not unto itself a problem, but if this is the case, I would strongly urge the authors to seek out guidance in this area. This is not meant as a derogatory comment – these concepts are challenging!

General scientific questions:

- (1) Precipitation due to the loss of the first twenty residues is intriguing, but I am unconvinced this is solely due to the electrostatics (two charged residues is still a very low charge density). If this is a narrative the authors wish to pursue, they should test a construct with D to A mutations at those two sites and demonstrate this too precipitates.
- (2) The text suggests droplets form at 35 degrees centigrade and dissolve at 15 degrees (this is lower critical solution temperature [LCST] behaviour, which should be noted), which begs the question, what happens between 15 and 25 degrees?! There should be a specific temperature phase boundary – i.e. given some concentration there must exist a temperature above which droplets form and below which they do not. This phase boundary should be at least qualitatively identified. Such a phase boundary can be identified by performing temperature ramp experiments at different concentrations of protein and noting the onset of droplet formation, although the rate of temperature change must be sufficiently slow.
- (3) What else does the CRD bind to? How does assembly (and disassembly via lactose or some other binding partner) relate to normal biological function (if at all?).
- (4) The sequence alignments suggest ~9 residue repeat motif that occurs multiple times. Some species possess more repeats than others. Any explanation for this?

Further questions and comments:

- (1) The authors make many claims about function, but true function is never actually assayed (galectin-3's primary function is not to agglutinate LPS micelles). The authors must tone down their claims of function given this is never tested.
- (2) Agglutination is never actually defined. This seems an important concept to identify!!
- (3) Solution turbidity should be quantified rather than showing pictures of Eppendorf tubes. This may be prohibitively difficult to do in retrospect (i.e. if experiments have been completed and repeating is challenging/expensive) so I would not say this is *necessary* for publication, but other's may disagree with my leniency here.
- (4) The authors claim that "*This is the first reported example of extracellular function being mediated by LLPS.*". Unfortunately, elastin is reasonably well established as a key extracellular matrix protein that assembles through phase separation (see the work of Fred Keeley, Ashutosh Chilkoti, and many, many others). This narrative will need to be changed, although could be simply slightly modified (e.g. "*first reported example that we are aware of in a non-structural context*"?).
- (5) The sequence analysis in figure 1 is nice; how do these properties (especially pi-pi interaction) change in the mutant sequences?
- (6) In the introduction page 3 lines 10-16 is a single sentence separated by multiple colons/semicolons. I would suggest the authors re-write this as a single sentence, and/or avoid colon/semi-colon use here.
- (7) Rather than referring to "hnRNPs", it is more appropriate to refer to hnRNPA1 and hnRNPA2 (the two proteins that have been studied in the context of phase separation). There are 20+ hnRNPs, the majority of which have not (yet) been studied.

- (8) Figure 3b is confusing to me. The WT sequence shows a clear temperature dependence based on the increase in temperature, indicative of hydrophobic interactions, and this is lost in the WY/G mutant (which makes sense). However, the effect on the loss of Trp (W/G) seems much more pronounced than upon the loss of Tyr (Y/G). Does this mean that those two Trp residues contribute substantially more than the eleven tyrosines (!?). If yes, this should be emphasized. A really powerful test of this (and again, may be beyond the scope of this paper) would be to replace those 11 tyrosine residues with tryptophan – would should, in principle, lead to *much* more robust phase separation, if not perhaps aggregation (due to the massive enhancement in intermolecular interactions).
- (9) The results stated that the WY/G NTD construct was transparent at 700mM NaCl concentrations in figure 3d but the corresponding panel is blank – please show this data.
- (10) No justification for the blank figure for WT NTD at 200mM is given in figure 3d. Was this condition transparent or did LLPS occur? The data in figure 2c suggests that the WT NTD is able to undergo LLPS at 1 mM NTD concentrations and NaCl concentrations between 100mM and 300mM, and as such I would expect the 200 mM NaCl condition for WT in panel 3d to display droplets. Some explanation of what is going on here is required – if this condition does not undergo LLPS, why?
- (11) It appears (from zooming in) that in Figure 3d for the 200 mM NaCl vs. W/G construct (bottom middle panel) that small droplets are observable. Is this correct? If yes, what does this mean (given that the WT NTD does not undergo phase separation at this protein/salt concentration)?
- (12) Page 5 line 29 *“We did observe condensates for the W/G construct after increasing the salt concentration to 2 M, but these condensates were smaller and were not all temperature-reversible.”* – please include the corresponding results in the supplementary information. The fact that the loss of just two tryptophan residues has such a pronounced impact on phase separation is interesting and should be noted in reference to the authors (excellent) previous paper [see ref: 35] and again perhaps explored (see point (7)).
- (13) Page 6 line 24 *“Full-length galectin-3 only undergoes LLPS when the protein concentration is higher than 2 mM and a certain amount precipitates (data not shown)”* – please show the data. The fact that full-length galectin-3 is so recalcitrant to phase separation further supports a model where phase separation is entirely dependent on higher-order assembly by LPS-presenting particles (e.g. micelles, bacterial cells etc.).
- (14) Page 7 line 24 *“because of the (large) size of the LPS micelles (Fig. 5d)”* – make clear the physical origins of this loss of signal (tumbling).
- (15) Page 7 line 32 *“The turbidity of the LPS/galectin-3 mixtures is proportional to the protein concentration (Fig. 5f), which is consistent with this velcro-like behavior of the NTD: the more galectin-3 molecules are attached to each micelle, the more they tend to aggregate”* – this is an important result; by quantifying the saturation concentration as a function of galectin-3 concentration, one can analytically ask if the saturation-concentration dependence behaves as would be expected for a colloid undergoing phase separation (where galectin-3 concentration becomes a proxy for the inter-molecular interaction strength of the colloid).
- (16) *“The inter- and intramolecular interactions between the NTD and the CRD can be distinguished using NMR peaks from the non-carbohydrate-binding site (colored blue in Fig. 1d; NMR spectra in Fig. 4d).”* – this is true, but it’s not clear from this description exactly how this is done? By mixing NTD and CRD and comparing with WT? In a single WT sample, NTD to non CRD binding site interactions could be either intra- or inter-molecular, although one would expect inter-molecular to be concentration dependent and intra- to not be (although this is not *necessarily* true – e.g. if intermolecular NTD interactions (NTD:NTD) reduce intra-molecular NTD:CRD interactions this would convolute any analysis done here).
- (17) Figure 5A and B, protein concentrations are not given in the results or methods section. Does LPS interaction with full length galectin-3 reduce the critical concentration for phase separation in comparison to wt-FL

galectin-3 in the presence of PEG? Although figure 5C-D makes a compelling argument for LPS micelles playing a key role in galectin LLPS, knowing whether some of the phase separation is driven by molecular crowding affects derived from the LPS concentrations vs actual protein localization to and agglutination driven by binding to the carbohydrate moieties would give more insight to the system and credence to the proposed model. This could also be tested by comparing NTD/CRD/FL with PEG (instead of LPS micelles). The expectation here would be loss of the CRD should enhance phase separation (in much the same way that NTD assembles more strongly than FL in the absence of PEG).

- (18) In this same vein, lactose is used to state that carbohydrate binding driven by the CRD's binding to the exposed glycans on micelles is necessary for galectin-3 phase separation. Could it also be the case that lactose binding causes a conformational change in the galectin-3 structure that renders it unable to undergo LLPS? Since full length galectin-3 phase separates in the presence of 10% PEG, incubation of FL galectin-3 and PEG with varying concentrations of lactose can rule out conformational changes that disrupt LLPS. Alternatively, if lactose + 10% PEG + full-length WT fails to phase separate, this suggests that the lactose-bound galectin-3 is unable to phase separate; i.e. that lactose both acts as a competitive inhibitor for the CRD binding to LPS, and that *also* after competing it diminishes the driving force for phase separation. I do not believe distinguishing between these two mechanisms is not critical for publication, but it would be great if possible and should be mentioned.
- (19) It appears that the LPS assays (in figure 5) were done in the absence of salt? This is a major departure to the preceding experiments, and makes comparison challenging. A micelle assay under equivalent (or at least physiologically relevant) salt concentrations would be beneficial.
- (20) Page 5 lines 11-27. This information may be readily known by some, but many readers will have to look up what different peak intensity ratios and changes in the transverse relaxation rate constants meant. A small primer on what/how these conclusions can be made from those changes would be of great benefit to a broader audience. Similarly, an explanation of what peak assignment means to provide context for the unassigned peaks is critical for those figures to be interpretable.
- (21) Page 10 line 3: "...of *super-enhancer aggregates in gene regulation*" super-enhancers are not aggregates – I'd suggest 'assemblies' instead.
- (22) I enjoyed the 'open questions' paragraph at the end – excellent end to the discussion.

Typos/errors etc

- (1) Page 4 line 22 – " The NTD sample condensates at 35 °C". "Condensates" should be "condenses".
- (2) Page 2 Line 8. Insert "A" or "Our" before "Lipopolysaccharide (LPS) micelle model shows....."
- (3) Figure 1 Line 16 and Figure 3 line 4 - "tyrosins" should be "tyrosines"
- (4) Figure 5 C. I find "Bad NMR Signal" a poor choice of words for the figure, something more precise would be appreciated.
- (5) Why does Figure 4c have some molecular structures on it? Seems out of place.

Reviewers' comments:

Reviewer #1 (Remarks to the Author):

In this study Chiu and colleagues investigate the oligomerization and liquid-phase separation of galectin-3 and its relation to galectin-3 agglutination. Galectin-3 is in this respect particularly interesting because it contains only a single carbohydrate-recognition domain, while other galectins contain tandem CRDs that mediate agglutination. Instead, galectin-3 contains an intrinsically disordered N-terminal domain, which has a sequence with prion-like properties and thus might undergo liquid-liquid phase separation. Overall this is a very interesting study that is summarized in a well-written and nicely illustrated manuscript.

We thank reviewer for this positive summary.

Major points:

LLPS is assessed by contrast microscopy and turbidity measurements. However, these are prone to artefacts (e.g. **formation of water droplets, non-LLPS oligomerization, ...**). Therefore it is important that LLPS of both the NTD and full-length galectin-3 is studied using fluorescent microscopy. In addition, the liquid-like properties of the droplets should be demonstrated by FRAP experiments.

We thank the reviewer for their suggestion to use fluorescent-labeled protein to demonstrate the liquid-like properties of our system. We constructed GFP-tagged protein and used fluorescence microscopy to demonstrate that the condensate we observed under the bright field microscope is indeed from the protein. We have also performed FRAP experiments to demonstrate the liquid-like properties of the droplets.

These results are shown in the panels we have added to Fig. 2c (NTD) and 2h (full length). The design of the construct, protein purification, and fluorescence microscopy experiments are described in the revised Methods section.

The results of the new FRAP experiments are consistent with our original results showing the LLPS properties of galectin-3. Because galectin-3 undergoes LLPS with LCST behavior (i.e. samples condense at high instead of low temperatures), it is unlikely that the droplets observed are water condensation. Furthermore, droplets were not observed under identical conditions for the CRD, WY/G-NTD, and WY/G-FL constructs, supporting the conclusion that the condensate is formed by the protein.

Furthermore, the data also show that condensation is reversible, with evidence of binodal crossing under various conditions (Fig 2a, 2g, and SI Movies) that is consistent with liquid-liquid phase separation behavior rather than aggregation (non-LLPS oligomerization). The observation of multiple fused droplets in the heterogeneous phase also supports the conclusion that the condensates have liquid-like properties.

LLPS data are mostly shown for the NTD and some small experiments are reported later in the manuscript for full-length galectin-3. More data (temperature/ionic strength dependence, sensitivity to aliphatic alcohols, fluorescence microscopy) should be provided for full-length galectin 3 and shown early on in the manuscript together with the data for the NTD.

We thank the reviewer for the suggestion to gather the full-length and NTD LLPS studies together; this makes the transition between paragraphs smoother. We have added data for full-length galectin-3 and for the NTD, as summarized in the

We have removed the results of the PEG-induced LLPS experiments from the revised manuscript (as per reviewer 2's suggestion). We have identified conditions under which full-length galectin-3 undergoes LLPS without PEG. The reason we present data on the LLPS of the NTD and the full-length protein is to demonstrate their multivalency and ability to form higher-order oligomers. The fact that full-length galectin-3 undergoes LLPS alone under non-physiological conditions has no biological meaning. The biological function is inferred from our lipopolysaccharide studies. The CRD acts as an anchor for the protein to dock on glycoproteins (or in our lipopolysaccharide assays), and the locally concentrated NTD agglutinates other micelles also coated with galectin-3, via the mechanism demonstrated in the LLPS experiments.

- Most/all NMR experiments/data reported in the manuscript were recorded under conditions, in which no LLPS occurs. This is a bit weird given that a major focus of the work is on LLPS. Thus, NMR experiments of NTD and full-length galectin-3 in conditions of LLPS should be included, in order to support the conclusions of the manuscript.

Our aim with the NMR spectroscopy and mutagenesis experiments was to investigate self-assembly. We do not think NMR studies are viable under these LLPS conditions because the condensates are too large to be detected by NMR, as shown in a previous study of ours on TDP-43 (Li et al., *BBA* 1866, 2018, 214). Under LLPS conditions, signals are only observed from the free monomer, not from the higher-order oligomer. Similar results have also been reported in a recent study of NPM1 (White et al., *Mol Cell* 74, 2019, 1): they recorded NMR spectra under non-LLPS conditions to identify the segments of the protein involved in LLPS; under LLPS conditions, signals from these segments are broadened. Furthermore, our LLPS conditions (especially for the full-length construct) involve high concentrations of NaCl, which are not suitable for NMR studies because the solutions become highly conducting (see Flynn et al., *JACS* 122, 2000, 4823, for example). Thus we combined solution-state NMR and mutagenesis data to show that the aromatic residues are critical for LLPS.

To demonstrate why the NMR and LLPS experiments were performed under different conditions, we measured for the NTD at 1 mM with 150 mM NaCl (the LLPS condition with the lowest NaCl concentration so that the spectrum obtained with the cytogenetic NMR probe remained of reasonable quality). See the inserted figure below, although the overall intensity decreases in the condensate present conditions (30 and 40 °C), the self-association is also enhanced (as described in the main text), resulting in the difference between intensity ratio and the sample molar ratio. However, we cannot differentiate the effect of the forming

higher-order oligomers and self-assembly. For example, at 40 °C, the intensity ratio between 1 mM and 0.1 mM sample is only 7.4; this could be resulted from the loss of signal of forming higher-order oligomers or from the enhanced self-association (because of hydrophobicity). Thus we only studied the self-association under one-phase condition. The sentences

“.....We did not apply the NMR studies on the sample in the heterogeneous state because the size of the condensate was too large for NMR detection and thus we cannot differentiate that the signal loss is because of forming higher-order oligomers or self-association. Furthermore, high salt concentration also deteriorates the quality of NMR spectrum.....” have also been added to explain why different conditions were used.

- Statements regarding “the first reported example of extracellular LLPS ...” should be removed; please see Kostylev et al. Mol Cell 2018.

We thank the reviewer for pointing this out. This sentence has been removed.

Reviewer #2 (Remarks to the Author):

The manuscript by Chiu et al. describes a mechanism by which galectin-3 can undergo liquid-liquid phase separation in extracellular space to mediate agglutination of lipopolysaccharide micelles. They utilize NMR to probe galectin-3 dynamics and inter- and intramolecular interactions between its N-terminal Domain and its Carbohydrate Recognition Domain. Solution turbidity assays are coupled with NMR to discern the amino acid residues that drive galectin-3 LLPS. They state that the galectin-3 CRD serves to increase local protein concentrations in the space between adjacent lipopolysaccharide micelles by binding to micelle carbohydrates and galectin-3’s disordered NTD drives phase separation of galectin-3 and subsequent agglutination of LPS micelles. The combination of high-resolution NMR with macroscopic phase separation phenomenon is appreciated and a strong component of the paper, as is the sequence analysis.

We thank the reviewer for these positive comments.

The general premise of this paper is highly intriguing, and I would say that I tend to think the author’s model may be partially if not entirely correct. However, my major concern in how the study is presented is the concentration regimes used of both protein and NaCl

used throughout this study (for different reasons!). I raise this as a concern, not because I believe this discounts the authors' model, but because instead I believe it sets up an incredibly simple 'straw man' that would be used to dismiss the authors' findings – namely that the conditions are so far from physiological relevance that the phenomena studied are simply an artefact of those conditions. At an absolutely minimum, a quantitative protein vs. salt phase diagram (the low-concentration arm of the binodal only) must be provided to allow the reader to understand the salt/protein concentration dependence. The authors hint at this in Figure 2e, but without actual data this sketch is insufficient to draw conclusions from. That said, I do believe an interesting narrative emerges from this work (hence the length of my review!), and I'd like to provide context for my concerns.

Firstly, why are the solution conditions problematic? Essentially every protein will undergo LLPS under the 'appropriate' conditions. Consequently, the observation of LLPS should not be considered evidence of biological relevance. For example, Figure 2a demonstrates the temperature dependence of 1 mM protein solution under semi-physiological salt conditions (150 mM NaCl). Even a small decrease in the concentration to 750 μ M (still 1-2 orders of magnitude higher than the protein concentration required for other systems studied in vitro) yields loss of phase separation (Figure 2c) suggesting this is below the (salt-dependent) saturation concentration. Figure 2d further makes the point that an extremely high concentration of protein and salt is required for LLPS to be observed. The salt dependence especially is concerning; given the remarkably small number of charged residues, **it seems more likely that the high salt concentration is driving a 'salting out' transition, as opposed to a screening of repulsive electrostatics.**

The authors then add 10% PEG-8000 (this is a considerably high concentration, given the more standard values are in the 0.5-2% range) which – as may be expected – decreases the saturation concentration into a more physiologically reasonable range. However, I have concerns regarding the molecular mechanism underlying the effect of PEG here. It is worth noting that much work (from Tom Record and others) has demonstrated that PEG cannot be treated as a purely excluded volume crowder, and there is a growing concern about the use of PEG in the context of phase separation studies. **I bring this up only to discourage the authors from using 10% PEG-8000 as a tool to "demonstrate" a physiological relevant concentration regime.**

However, the authors do then demonstrate that that full-length WT protein + LPS at a protein concentration of 40 μ M protein, a much more reasonable protein concentration, but with (seemingly) no additional NaCl beyond the PBS buffer? However, assuming this result is robust to the absence of NaCl, This for me, and specifically Figure 5a, is the most interesting result – a protein that fundamentally does not undergo LLPS particularly strongly will robustly do so in the presence of its binding partner in a manner that is dependent on both the disordered N-terminal domain and the LPS binding CRD.

With this in mind, I would strongly recommend the authors consider a slightly alternative narrative to the one presented. **Explicitly make**

the point that while monomeric galectin-3 can undergo phase separation and that phase separation appears to be dependent on the Trp and Tyr residues, highly non-physiological conditions are required. This allows the authors the opportunity to dissect that sequence-determinants of phase behaviour, while recognizing that the conditions are somewhat ridiculous. In this, a true protein/salt phase diagram would be necessary for this to be a sufficiently rigorous study. Having then established that phase separation can occur, but requires these somewhat extreme solution concentrations, the authors can then ask if this behaviour is simply a quirk of the underlying physical chemistry, or might there be functional relevance to this behaviour?

From this the authors can introduce the LPS micelles and propose the model that the micelles provide the requisite higher-order multi-valency to drive phase separation. In effect, the galectin-3 does not (readily) undergo phase separation, instead, micelles coated by galectin-3 (where this coating is dependent on the ability of galectin-3 to bind LPS) are truly the basic units that undergo phase separation. In this model, galectin-3 only undergoes phase separation after forming a supramolecular assembly of LPS micelle coated in galectin-3. The authors should calculate the (theoretical) local concentration of galectin-3 bound to the surface of the LPS micelles and use this to compare with real bacterial to ask if this is (putatively) a relevant biological mechanism. I should say, this is basically the same narrative as the authors have now; the key difference in my opinion is explicitly recognizing that in the absence of a supramolecular assembly phase separation of galectin-3 requires entirely non-physiological conditions to occur [BUT this still allows a dissection of the relevant molecular interactions].

In short, the study here does not, in my opinion, provide compelling evidence that galectin-3 (by itself) undergoes phase separation under physiological conditions, which is 100% OK! However, it does suggest that if galectin-3 can be locally concentrated (e.g. on the surface of bacteria) this may enable the saturation concentration for phase separation to become sufficiently low. It's not clear to me if agglutination can actually be described using phase separation as a physical framework (even just in vitro). To argue that galectin-3 drives agglutination through phase separation would require the demonstration that in vitro agglutination shows typical hallmarks of phase separation (e.g. a saturation concentration). This may not be technically possible to achieve, in which case the narrative should state that galectin-3 can undergo phase separation under very high protein/salt concentrations, and those same interactions drive agglutination in a manner that is essentially dependent on the recruitment of galectin-3 to the surface of LPS micelles (or to gram-negative bacteria). This is not a problem and does not undermine the study – a functional two-phase system is not the only interesting mode of assembly!

What I don't know is if this model makes any kind of biological sense and it is the author's responsibility to push this further – i.e. in the density of LPS on the surface of gram-negative bacteria sufficiently high to drive assembly in a manner analogous to that observed for LPS micelles? This most explicit test of this would be to

ask if the WY/G version is able to drive agglutination of some gram negative bacteria (e.g. E. coli - the expectation being no) but I fully recognize this is likely beyond reasonable scope for revisions in this manuscript.

An alternative explanation is that phase separation is an unavoidable consequence of the galectin NTD being sufficiently multivalent to interact with (and modulate?) the behaviour of the CRD. It is difficult to rule this mechanism out (or in) without disrupting the phase behaviour and performing in vivo studies (which goes beyond the reasonable scope of this paper) but **the possibility of phase separation here being simply an epiphenomenon should be mentioned in the discussion.**

Finally, there is language in the paper that makes me wonder if the authors fully understand the underlying physical concepts surrounding a two-phase equilibria (i.e. binodal curves, saturation concentrations, the equalization of chemical potential across the two phases etc.). This is not unto itself a problem, but if this is the case, I would strongly urge the authors to seek out guidance in this area. This is not meant as a derogatory comment - these concepts are challenging!

We are very gratefully for the reviewer's comments. Before we start to answer the questions point-by-point, we summarize what we have changed in the manuscript based on the reviewer's suggestions and comments.

(1) We have changed the overall narrative of the manuscript:

We introduce LPS micelles as model tools to promote galectin-3 agglutination and highlight the long-standing question of how this monomeric protein uses multivalency to agglutinate (revised Fig. 1). At this point, we mention how LPS or different cell types have been used in a number of studies as models to study galectin-3's agglutination (page 4, at the end of the Introduction and the beginning of the Results; related references are included). Although it is well-established that the presence of the NTD is critical for agglutination, the mechanism involved is unknown. Our data show that the NTD and full-length galectin-3 both undergo LLPS (Fig. 2). The NMR and mutagenesis data show that multivalent connections are mediated by aromatic residues (Figs 3 & 4). The phase diagram shows that high, non-physiological salt and protein concentration are required for LLPS but we explain that this is not a problem for interpreting galectin-3's agglutination function. We use the LPS assays and NMR data from different constructs to demonstrate that the binding of the CRD on the surface of the LPS micelles increases the local concentration of the NTD (Fig. 5). The accumulation of weak interactions between the locally concentrated NTDs leads to agglutination, via a mechanism similar to the one that drives LLPS. Finally, we show that the interactions between galectin-3 and the LPS micelles are transient

and dynamic. NMR data recorded in the presence of lactose show that there are no stable oligomers in the dissolved galectin-3/LPS aggregates. Galectin-3 remains monomeric (Fig. 6).

(2) We have removed all the PEG-related results:

We are especially grateful for this suggestion to avoid using PEG. Although this crowding agent is often used in LLPS-related studies, especially to “induce LLPS under physiological conditions”, demonstrating that galectin-3 undergoes LLPS under physiological conditions is not our objective here. Indeed, galectin-3 does not condense by itself intra- or extracellularly, and our main focus is on the functional role of agglutination (requiring μM -range samples and anchoring bodies such as glycoproteins or LPS). This is why we have removed the investigations involving PEG from the manuscript.

Our original idea was to show that while full-length galectin-3 condenses in PEG the WY/G construct does not. To demonstrate the role in LLPS of the aromatic residues in full-length galectin-3, in support of our NMR and mutagenesis results (Fig. 4), we searched for conditions under which full-length galectin-3 undergoes LLPS. Our criterion for determining that the condensates that form come from liquid-liquid phase separation rather than aggregation was the observation of LCST phase behavior. Galectin-3 precipitated at concentrations above $600 \mu\text{M}$ (these aggregates did not show LCST behavior under microscopic observation), but LCST phase behavior was observed in a $500 \mu\text{M}$ sample with more than 700 mM NaCl (revised Fig. 2f,g). Under the same conditions, the WY/G full-length construct no longer undergoes LLPS (new Fig. 4j).

(3) The fact that the wild-type NTD and full-length galectin-3 show LCST behavior but the aromatic-residue-removed and CRD-only constructs do not suggests that “salting-out” is not the main phenomenon involved. The only two negatively charged residues in the NTD induce electrostatic repulsion. This force can be compensated by enhanced hydrophobicity (increased temperature) or higher ion concentration.

(4) We have toned down the functional interpretations and mention the possibility that the LLPS of the NTD is an epiphenomenon in the discussion (Page 11).

(5) The functional importance of the disordered NTD has been extensively studied and is well-established. However, how the disordered domain fulfills its function is unknown. We have added citations and a couple of sentences in the

Introduction and Discussion sections outlining the biological function of galectin-3 (highlighted in red).

(6) We have studied the textbook Polymer Physics (Rubinstein & Colby, 2003) and a few theoretical papers (e.g. Posey et al. 2018, *Methods Enzymol*) to ensure that we describe the observed phenomenon more precisely. We hope any ambiguities or misleading statements in the descriptions of the phase transitions have been removed from the revised version of the manuscript (changes highlighted in red).

(7) In light of the extensive revisions, we have changed the title of the article from “Liquid-liquid phase separation mediates extracellular function through the accumulation of multivalent interactions” to “Liquid-liquid phase separation and multivalent interactions in extracellular protein function: the tale of galectin-3”.

General scientific questions:

(1) Precipitation due to the loss of the first twenty residues is intriguing, but I am unconvinced this is solely due to the electrostatics (two charged residues is still a very low charge density). If this is a narrative the authors wish to pursue, they should test a construct with D to A mutations at those two sites and demonstrate this too precipitates.

We agree with this comment. Since the role of the charged residues is not the focus of the article, we have removed this statement for the sake of clarity.

The fact that the negative charges are conserved in vertebrates is of itself an interesting topic. We are investigating charge effects by replacing the charged residues and by creating phosphomimetic mutants (Ser-6 and Ser-12 are reported phosphorylation sites). Our working hypothesis is that these two negatively charged residues, and the level of phosphorylation, regulate both inter- and intramolecular interactions, and thus galectin-3's tendency to form higher-order oligomers.

(2) The text suggests droplets form at 35 degrees centigrade and dissolve at 15 degrees (this is lower critical solution temperature [LCST] behaviour, which should be noted), which begs the question, what happens between 15 and 25 degrees?! There should be a specific temperature phase boundary – i.e. given some concentration there must exist a temperature above which droplets form and below which they do not. This phase boundary should be at least qualitatively identified. Such a phase boundary can be identified by performing temperature ramp experiments at different concentrations of protein and noting the onset of droplet formation, although the rate of temperature change must be sufficiently slow.

We are grateful for these suggestions. This LCST behavior is now mentioned in the revised manuscript. We have performed temperature ramp experiments to define the phase boundary (an example of how we determine the phase boundary is shown in Fig. 2a, and the general trend is shown in Fig. S1). The binodal temperature under the conditions used for Fig. 2a is 19 °C. Above this temperature, nucleation occurs (small condensates are observed). These condensates grow as the temperature is increased (at each temperature, we allowed the samples to equilibrate for 3 min before collecting the micrograph).

(3) What else does the CRD bind to? How does assembly (and disassembly via lactose or some other binding partner) relate to normal biological function (if at all?).

The CRD of galectin-3 binds to many glycoproteins and glycolipids (many of which are immune cell receptors) via their canonical carbohydrate binding site (reviews: Rabinovich and Toscano, *Nature* 9, 2009, 338-352; Johannes et al. *J Cell Sci* 131, 2018, 1-9). Interestingly, many non-glycoproteins, notably intracellular proteins, also interact with the CRD of galectin-3, e.g. Bcl-2, CBP70, Gemin4, (review: Haudek et al. *BBA*, 2010, 189-191). How these proteins without a sugar moiety interact with galectin-3 remains unknown. Given that we have shown that the NTD can interact through the non-carbohydrate binding site, it is possible that many other proteins also interact with the same site and thereby compete with the NTD-CRD interaction. For many proteins furthermore, it remains unclear which domain (NTD or CRD) they interact with (e.g. hnRNP Q, K-Ras, and Synexin). The NTD also interacts with other proteins, for example, the C-terminal domain of Alix (ALG-2-interacting protein X) involved in HIV-1 budding (Wang et al. *Glycobiology* 24, 2014, 1022). PONDR predictions indicate that the Alix-interacting domain is also intrinsically disordered. Many studies have also demonstrated galectin-3's role in gene regulation and RNA interactions (Patterson et al. *Methods Mol Biol* 1207, 2015, 431-49), leading to the hypothesis that the disordered NTD may play the same role as the IDRs of other hnRNPs such as hnRNPA1, TDP-43, and FUS, namely forming the higher-order oligomers involved in gene regulation functions. Would the binding of another protein to the non-carbohydrate binding site release NTD and enhance LLPS? Or as the reviewer points out below (point 18), does binding to the carbohydrate-binding site cause conformational change that also regulates multivalency? How these complicated networks regulate the interactions between the NTD and the CRD and the formation of higher-order oligomers are questions we are looking into right now (as discussed in the final paragraph).

(4) The sequence alignments suggest ~9 residue repeat motif that occurs multiple times. Some species possess more repeats than others. Any explanation for this?

This is an interesting question but we have no explanation to offer. A naive guess is that it may be related to body temperature (because of the LCST behavior) but there is no clear trend. In addition to the number of aromatic residues, other factors may also be involved in controlling the multivalent connection. We think that NTD/CRD interactions are also important in regulating multivalency. Apart from the number of repeats, the charged residues and the charges on the non-carbohydrate binding sites (the blue bars in Fig. 3a) may also be determinants for the ability to undergo LLPS. The table below gives a few examples: in zebrafish, there are fewer aromatic residues in the NTD but more negative charges; compared with other species, the non-carbohydrate binding site (nonCBS) in zebrafish also has more negatively charged residues. These may repel the NTD away from the nonCBS (and populate the state shown in Fig. 1d instead of the one in Fig. 1c).

	W	Y	W+Y	NTD -	NTD +	NTD Net	nonCBS -	nonCBS +	nonCBS Net
Human	2	10	12	2	0	-2	1	3	2
Rabbit	2	9	11	2	0	-2	0	3	3
Rat	2	10	12	2	0	-2	0	3	3
Dog	2	13	15	2	0	-2	1	3	2
Pig	2	11	13	2	1	-1	0	3	3
Bovine	2	12	14	2	1	-1	0	3	3
Chicken	1	9	10	3	1	-2	0	3	3
Frog	2	8	10	6	2	-4	2	3	1
Zebrafish	6	1	7	4	0	-4	6	3	-3

Further questions and comments:

(1) The authors make many claims about function, but true function is never actually assayed (galectin-3's primary function is not to agglutinate LPS micelles). The authors must tone down their claims of function given this is never tested.

We have rewritten large parts of the discussion. We have removed several passages in which we offered explanations based on our model of the findings of

previous studies and have toned down our interpretations. We moved what was Fig. 6, an explanation of known galectin-3 functions based on our model, to the supplementary information (Fig. S9; this figure remains included to show how our model can explain these functions instead of the pentamer model used elsewhere in the literature to explain galectin-3's multivalency, e.g. Lau et al. *Cell* 129, 2007, 123-134; Goetz et al. *J Cell Biol* 180, 2008, 1261-75).

(2) Agglutination is never actually defined. This seems an important concept to identify!!

We thank the reviewer for this suggestion. We have added a sentence on Page 4 Line 3 “.....One of galectin-3's functions is to agglutinate organelles, such as neutrophils and laminin, or glycosylated molecules. This function, in which multivalent galectin-3 acts as a “bridge”, is lost when the NTD is removed.....”. This is what the term “agglutination” refers to in the article.

(3) Solution turbidity should be quantified rather than showing pictures of Eppendorf tubes. This may be prohibitively difficult to do in retrospect (i.e. if experiments have been completed and repeating is challenging/expensive) so I would not say this is necessary for publication, but other's may disagree with my leniency here.

We have repeated the turbidity experiments. The results from triplicated experiments are now provided in the revised Figs 1i, 5a, and Tables S1, S2, and S3.

(4) The authors claim that “This is the first reported example of extracellular function being mediated by LLPS.”. Unfortunately, elastin is reasonably well established as a key extracellular matrix protein that assembles through phase separation (see the work of Fred Keeley, Ashutosh Chilkoti, and many, many others). This narrative will need to be changed, although could be simply slightly modified (e.g. “first reported example that we are aware of in a non-structural context?”).

We thank the reviewer for pointing out these studies. We have removed the entire sentence in question here, which was not important for the message of the corresponding paragraph.

(5) The sequence analysis in figure 1 is nice; how do these properties (especially pi-pi interaction) change in the mutant sequences?

This is a good suggestion. The WY/G and Y/G mutants do change the predicted pi-pi interactions (revised Fig. 4i). The level of structural disorder and prion-likeness does not change in these aromatic-replaced mutants (Fig. S7).

(6) In the introduction page 3 lines 10-16 is a single sentence separated by multiple colons/semicolons. I would suggest the authors re-write this as a single sentence, and/or avoid colon/semi-colon use here.

These sentences have been revised.

(7) Rather than referring to “hnRNPs”, it is more appropriate to refer to hnRNPA1 and hnRNPA2 (the two proteins that have been studied in the context of phase separation). There are 20+ hnRNPs, the majority of which have not (yet) been studied.

Thank you for pointing this out. We now refer specifically to hnRNP A1 and A2 in the revised version instead of hnRNPs in general (Line 11, Page 3).

(8) Figure 3b is confusing to me. The WT sequence shows a clear temperature dependence based on the increase in temperature, indicative of hydrophobic interactions, and this is lost in the WY/G mutant (which makes sense). However, the effect on the loss of Trp (W/G) seems much more pronounced than upon the loss of Tyr (Y/G). Does this mean that those two Trp residues contribute substantially more than the eleven tyrosines (!?). If yes, this should be emphasized. A really powerful test of this (and again, may be beyond the scope of this paper) would be to replace those 11 tyrosine residues with tryptophan – would should, in principle, lead to much more robust phase separation, if not perhaps aggregation (due to the massive enhancement in intermolecular interactions).

We do not think it is possible to conclude that W is more important than Y based on these NMR data. In fact, it is the intensity ratio for the Y/G mutant (not the W/G mutant) that is 0.1 at 10 °C. Our hypothesis is that since W is more hydrophobic than Y (according to most amino-acid scales), once it is removed, the temperature-dependence is reduced but the remaining Ys still contribute to self-association (thus the ratio is no 0.1). Furthermore, peak intensities in an HSQC spectrum depend on a combination of many factors. The inter- and intramolecular conformational equilibrium may cause differential line broadening because of chemical exchange (as illustrated in Fig. S4; the levels of exchange for the W/G and Y/G mutants are expected to differ because they do not have the same number of aromatic residues). Thus, as mentioned in the text, because there is not an equal number of tyrosine and tryptophan residues, our data do not indicate which amino-acid type is more important. However, we do think the reviewer’s suggestion (compare 12 W with 12 Y) would be a very powerful way to work this out. This may also explain why zebrafish galectin-3 has 6 W and 1 Y while in other vertebrates Y predominates (see table above). We fear adding these explanations would lead the reader off on a tangent though, so we would rather leave this part of the text unchanged.

(9) The results stated that the WY/G NTD construct was transparent at 700mM NaCl concentrations in figure 3d but the corresponding panel is blank – please show this data.

Comments **(9) to (12)** seemed to arise from a misunderstanding of the original version of Fig. 3d. We have moved this to Fig. S5 and for the sake of clarity, we have simplified Fig. 3e to only show the data required to justify our conclusion. This misunderstanding may have been caused by the layout of the panels (which were aligned, perhaps unnecessarily, according to protein and salt concentrations) and a misreading of the units (a common feature of the following questions is a switching of 2000 mM and 200 mM NaCl).

Specifically regarding question (9): we do not mention the 1 mM WY/G sample in 700 mM salt in the text. We directly observed this construct in a high salt condition (because under these conditions, the W/G mutant still condenses at 30 °C)

(10) No justification for the blank figure for WT NTD at 200mM is given in figure 3d. Was this condition transparent or did LLPS occur? The data in figure 2c suggests that the WT NTD is able to undergo LLPS at 1 mM NTD concentrations and NaCl concentrations between 100mM and 300mM, and as such I would expect the 200 mM NaCl condition for WT in panel 3d to display droplets. Some explanation of what is going on here is required – if this condition does not undergo LLPS, why?

The concentration discussed in this part of the text is 2000 mM , not 200 mM NaCl. In the presence of 150 mM NaCl, 1 mM wild-type NTD undergoes LLPS (Fig. 2), and therefore we did not perform experiments on 1 mM NTD in 2 M NaCl. We have changed the units and layout of the figures to avoid this confusion.

(11) It appears (from zooming in) that in Figure 3d for the 200 mM NaCl vs. W/G construct (bottom middle panel) that small droplets are observable. Is this correct? If yes, what does this mean (given that the WT NTD does not undergo phase separation at this protein/salt concentration)?

Here, the concentration is 2000 mM NaCl (not 200 mM). We did observe condensation for the W/G mutant with 2 M salt, presumably because of the contributions of the tyrosine residues. There was no condensation however for the WY/G mutant under the same conditions (1 mM protein concentration and 2 M NaCl; revised Fig. 3e).

(12) Page 5 line 29 “We did observe condensates for the W/G construct after increasing the salt concentration to 2 M, but these condensates were smaller and were not all temperature-reversible.” – please include the corresponding results in the supplementary information. The fact that the loss of just two tryptophan residues has such a pronounced impact on phase separation is interesting and should be noted in reference to the authors (excellent) previous paper [see ref: 35] and again perhaps explored (see point (7)).

These results did appear in the original version and are mentioned in comment (11). As mentioned above, the layout of the figure has been altered (Fig. S5) to avoid ambiguity and we only highlight the effect for the WY/G mutant (revised Fig. 3e). These data are insufficient to conclude that W is more important than Y (see our reply to comment (8))

(13) Page 6 line 24 “Full-length galectin-3 only undergoes LLPS when the protein concentration is higher than 2 mM and a certain amount precipitates (data not shown)” – please show the data. The fact that full-length galectin-3 is so recalcitrant to phase separation further supports a model where phase separation is entirely dependent on higher-order assembly by LPS-presenting particles (e.g. micelles, bacterial cells etc.).

We noticed that full-length galectin-3 has a strong tendency to aggregate and our original idea was to highlight the difficulty of using the full-length protein to achieve LLPS, thus we chose to use PEG to achieve our purpose. However, following the reviewer’s suggestion to not use PEG, we have extensively studied full-length galectin-3’s LLPS ability (see above, and Fig. 2) and have identified conditions under which galectin-3 undergoes LLPS (with LCST phase behavior) but does not aggregate. As the reviewer mentions, this resistance to phase separation supports our LPS agglutination model because a locally increased concentration is required.

(14) Page 7 line 24 “because of the (large) size of the LPS micelles (Fig. 5d)” – make clear the physical origins of this loss of signal (tumbling).

This sentence has been changed to “.....and thus those CRDs’ NMR signal is undetectable because of the reduced overall tumbling rate of the protein on the large LPS micelles.....”

(15) Page 7 line 32 “The turbidity of the LPS/galectin-3 mixtures is proportional to the protein concentration (Fig. 5f), which is consistent with this velcro-like behavior of the NTD: the more galectin-3 molecules are attached to each micelle, the more they tend to aggregate” – this is an important result; by quantifying the saturation concentration as a function of galectin-3 concentration, one can analytically ask if the saturation- concentration dependence behaves as would be expected for a colloid undergoing phase separation (where galectin-3 concentration becomes a proxy for the inter-molecular interaction strength of the colloid).

This is a good suggestion, but our methods would not be sufficiently quantitative to demonstrate transition or saturation. The new Fig. 1h shows results of tests for a range of LPS and galectin-3 concentrations. The transition occurs at between 5 and 10 μ M protein. However, the standard deviation of each independently

prepared and measured sample is too large to properly quantifying the transition point (Table S1). In addition, when the concentrations of protein and LPS are both high, the optical density reaches the limit of our spectrophotometer, making it difficult to determine whether the saturation concentration has been reached. Other analytical techniques such as dynamic light scattering might be more suitable to answer this question. To avoid any ambiguity, we only use the term “agglutination” in the discussion of the LPS/galectin-3 results.

Indeed, agglutination does not necessarily involve LLPS as in other dimer or tandem-repeated galectins. In this study, we show that agglutination proceeds through the accumulation of weak multivalent interactions between aromatic residues. These multivalent interactions involved in forming high-order oligomers resemble the mechanism that drives LLPS for the NTD and full-length galectin-3. This is reminiscent of the intrinsically disordered FG-repeats of nucleoporins which are locally concentrated in the nuclear pore transport complex. These repeats are also regarded as biomolecular condensates (Banani et al. *Nature Reviews* 18, 2017, 285) and their phase properties have been studied (Schmidt and Gorlich, *Trends Biochem Sci* 41, 2016, 46-61). A recently published example of this effect is Ries et al.’s study of the m6A-binding protein DF2 (Ries et al. *Nature* 571, 2019, 424): DF2 interacts weakly with methylated mRNA. DF2 only functions when mRNA is polymethylated, thereby recruiting many copies of DF2 such that the IDR of DF2 becomes concentrated enough to undergo LLPS. In contrast, singly methylated mRNA does not undergo LLPS (as illustrated by Ries et al. in Fig. 7 of the Extended Data). These two examples are now mentioned in the Discussion.

(16) “The inter- and intramolecular interactions between the NTD and the CRD can be distinguished using NMR peaks from the non-carbohydrate-binding site (colored blue in Fig. 1d; NMR spectra in Fig. 4d).” – this is true, but it’s not clear from this description exactly how this is done? By mixing NTD and CRD and comparing with WT? In a single WT sample, NTD to non CRD binding site interactions could be either intra- or inter- molecular, although one would expect inter-molecular to be concentration dependent and intra- to not be (although this is not necessarily true – e.g. if intermolecular NTD interactions (NTD:NTD) reduce intra- molecular NTD:CRD interactions this would convolute any analysis done here).

In our previous study (Lin et al. *JBC* 292, 2017, 17845), we used different lengths of NTD-truncated forms of galectin-3 (with the first 10, 20, 30...etc. residues removed) and intermolecular paramagnetic relaxation enhancement NMR methods to demonstrate that the NTD interact fuzzily with the non-carbohydrate binding site of the CRD inter- and intramolecularly. We also showed that there

are intermolecular interactions between NTDs. To make this clearer, we have added a paragraph “.....The chemical shift differences between the CRD-only and full-length constructs indicate that the NTD and CRD interact intramolecularly via the non-carbohydrate-binding site (blue arrows in Fig. 4d; schematic representation on the right). The concentration-dependent chemical shift perturbations in the non-carbohydrate-binding site for wild-type galectin-3 indicate that the NTD and CRD also interact intermolecularly (orange arrows in Fig. 4d; see ref. ⁴⁹ for more details).....” and a diagram on the right of the revised Fig. 4.

(17) Figure 5A and B, protein concentrations are not given in the results or methods section. Does LPS interaction with full length galectin-3 reduce the critical concentration for phase separation in comparison to wt-FL galectin-3 in the presence of PEG? Although figure 5C-D makes a compelling argument for LPS micelles playing a key role in galectin LLPS, knowing whether some of the phase separation is driven by molecular crowding affects derived from the LPS concentrations vs actual protein localization to and agglutination driven by binding to the carbohydrate moieties would give more insight to the system and credence to the proposed model. This could also be tested by comparing NTD/CRD/FL with PEG (instead of LPS micelles). The expectation here would be loss of the CRD should enhance phase separation (in much the same way that NTD assembles more strongly than FL in the absence of PEG).

The sample conditions are now indicated in the results and described in the methods.

Because all PEG-related results have been removed in the revised version, we make no attempt to differentiate crowding effects from those of the protein binding to the surface of the LPS micelles. However, we did test the effect of PEG on the NTD as suggested by the reviewer. Unexpectedly (at least for us), PEG does not make NTD condense at a lower concentration. See the inserted figure **a**, at a condition that 100 μ M full-length galectin-3 shows condensates with PEG (in the old version Fig 4), but the 100 μ M sample of NTD with 10% PEG remains homogeneous. Moreover, the presence of PEG changes the NTD's phase transition behavior from LCST to UCST (see the inserted figure **b**): the sample is more turbid at lower temperatures than at higher ones. We are therefore grateful for the reviewer's warning about using PEG, which echoes those mentioned in the literature (e.g. Alberti et al, *Cell* 176, 2019, 419). PEG leads to artifacts in our system and we have therefore removed the

corresponding results from the revised manuscript.

(18) In this same vein, lactose is used to state that carbohydrate binding driven by the CRD's binding to the exposed glycans on micelles is necessary for galectin-3 phase separation. Could it also be the case that lactose binding causes a conformational change in the galectin-3 structure that renders it unable to undergo LLPS? Since full length galectin-3 phase separates in the presence of 10% PEG, incubation of FL galectin-3 and PEG with varying concentrations of lactose can rule out conformational changes that disrupt LLPS. Alternatively, if lactose + 10% PEG + full-length WT fails to phase separate, this suggests that the lactose-bound galectin-3 is unable to phase separate; i.e. that lactose both acts as a competitive inhibitor for the CRD binding to LPS, and that also after competing it diminishes the driving force for phase separation. I do not believe distinguishing between these two mechanisms is not critical for publication, but it would be great if possible and should be mentioned.

This is a very good point. Instead of using PEG, we investigated the effects of lactose at conditions under which full-length galectin-3 undergoes LLPS (500 μ M protein, 1 M NaCl, 30 °C). The sample became less turbid when 25 mM lactose was added (micrographs on the right-hand side). The reviewer's hypothesis that the binding of lactose causes a conformational change that disrupts phase separation is consistent with the chemical shift changes observed on the non-carbon-hydrate

binding side (i.e. the NTD binding side; new Fig. 6d). There are several potential explanations for this. For example, since the LLPS of full-length galectin-3 involves NTD-CRD interactions (as shown in the NMR data, e.g. Fig. 4) in addition to NTD-NTD interactions, the conformational change induced by lactose binding may hinder these NTD-CRD interactions and thus lower the threshold for phase separation. How the inter- and intramolecular NTD/CRD interactions regulate LLPS is indeed a question we are looking into (as mentioned at the end of the conclusion paragraph). In the revised manuscript, we mention this possibility (“.....We cannot rule out the possibility that interactions between lactose and the CRD also cause a conformational change that reduces the propensity to phase separate as there were chemical shift perturbations in the non-carbohydrate-binding site (Fig. 6d).....”).

(19) It appears that the LPS assays (in figure 5) were done in the absence of salt? This is a major departure to the preceding experiments, and makes comparison challenging. A micelle assay under equivalent (or at least physiologically relevant) salt concentrations would be beneficial.

We tested the turbidity of LPS samples with 300 mM NaCl at different protein concentrations. The effect is insignificant (Fig. S8).

(20) Page 5 lines 11-27. This information may be readily known by some, but many readers will have to look up what different peak intensity ratios and changes in the transverse relaxation rate constants meant. A small primer on what/how these conclusions can be made from those changes would be of great benefit to a broader audience. Similarly, an explanation of what peak assignment means to provide context for the unassigned peaks is critical for those figures to be interpretable.

We have added brief explanations of how changes in peak intensities and relaxation rates are interpreted “.....indicating that the signals in the spectra from the 400 μ M are broader, presumably because of self-association.....”; “.....which reflect the overall effects of tumbling, the internal motions of the molecule, and chemical exchange due to micro-to-millisecond timescale motion.....”. The peak assignments and the fact that there are unassigned peaks (u.a.) are also explained in the text for a general audience. “.....We mapped the NMR chemical shift assignments of the wild-type and the W/G construct based on a previous publication. We did not assign the chemical shifts of the Y/G and WY/G constructs (marked as unassigned in Fig. 3) because only the overall trend in the peak intensity ratios and R_2 changes are critical to our interpretation. Furthermore, severe overlap in the NMR spectra of these two constructs (because of the many repeated amino-acid motifs) would have made assignment difficult. Overlapping peaks also provide limited residue-specific information. In these constructs, we therefore only analyzed the well-resolved peaks to avoid bias due to peak overlap.....”

(21) Page 10 line 3: “...of super-enhancer aggregates in gene regulation” super-enhancers are not aggregates – I’d suggest ‘assemblies’ instead.

This example has been removed because we have toned down the functional interpretations as suggested in comment (1).

(22) I enjoyed the ‘open questions’ paragraph at the end – excellent end to the discussion.

We thank this comment. Indeed, in recent literatures, galectin-3 is reported mostly as a biomarker. However, there are still many unknown properties of this protein that are critical to its function. Here we hope we have started a new framework for studying its function. Many are await for further investigation.

Typos/errors etc

(1)Page 4 line 22– “The NTD sample condensates at 35°C”. “Condensates” should be “condenses”.

(2)Page 2 Line 8. Insert “A” or “Our” before “Lipopolysaccharide(LPS) micelle model shows.....”

(3) Figure 1 Line 16 and Figure 3 line4 - "tyrosins" should be "tyrosines"

We thank the reviewer for carefully reading our manuscript. These typos have been corrected.

(4) Figure 5 C. I find "Bad NMR Signal" a poor choice of words for the figure, something more precise would be appreciated.

We have replaced the phrase by "Poor S/N ratio".

(5) Why does Figure 4c have some molecular structures on it Seems out of place.

These were graphical representations of the CRD and the WY/G mutant (no NTD-CRD interactions). We have removed them to make the figure clearer.

Reviewer #3 (Remarks to the Author):

This manuscript builds on previous observations by the authors of LLPS in galectin-3, a lectin with a disordered, low-complexity N-terminal domain and a C-terminal carbohydrate binding domain. Here, the authors use NMR spectroscopy and microscopy to characterize the influence of salt, temperature and protein concentration on LLPS in this system, as well as to demonstrate the role of aromatic residues in driving LLPS of the N-terminal domain. They also provide some initial data suggesting a role for LLPS in the biological functions of galectin-3, by providing a mechanism for multivalency and agglutination, in the absence of multiple CBDs or an oligomeric protein.

This is in general a well written manuscript that provides new insight, particularly into the mechanism of galectin-3 LLPS. The experiments are well carried out, and the link between the data and the conclusions is clear. It adds to the growing story of lectin LLPS, and also provides a new example of a functional extracellular protein LLPS, for which there are relatively few examples known so far. There are, however two areas in which additional detail or experiments would allow for more concrete conclusions to be drawn, strengthening the manuscript.

We thank the reviewer for these positive comments.

1. It is not clear why large sections of some constructs are listed as unassigned. For example, the HSQC data for the Y/G and WY/G NTD constructs look well-resolved, and have favorable R2 values, yet are unassigned. If there are other factors at play, the authors should specify them, and if not, then complete assignment can only improve the data presented. Without sequential resonance assignments, then plotting the R2 or delta-chemical shift data for these constructs as a comparison to assigned data sets is not particularly useful, rather only the aggregate or average values hold meaning. Complete assignments might also provide some additional site-specific information regarding preferred sites for inter/intramolecular interaction.

The wild type and the W/G mutant have many inter- and intramolecular interactions that equilibrate on slow or intermediate timescales (as illustrated in Fig. S4). Slow exchange leads to additional peaks in NMR spectra (see the chemical shift assignments, BMRB code 19491) and intermediate exchange leads to line broadening. The published assignments can be used to analyze the data from wild-type galectin-3 and the W/G mutant (there are only two substitutions in the latter and most of the peaks can be transferred from the wild type assignment). The WY/G and Y/G NTD HSQC spectra look well-resolved because the inter- and intramolecular interactions are suppressed. However, there are many similar and repeated motifs in these constructs that make the HSQC spectra very crowded and many peaks overlap. This makes sequential assignment very difficult. More importantly, since we are only interested in the overall change in the NMR parameters (intensity ratio and R_2) for these constructs, including the assigned W/G mutant and the wild type, average values are all that are needed for our interpretation (as shown in the panels on the right of Fig. 3c). To make this point more clearly, we have added “.....We mapped the NMR chemical shift assignments of the wild-type and the W/G construct based on a previous publication. We did not assign the chemical shifts of the Y/G and WY/G constructs (marked as unassigned in Fig. 3) because only the overall trend in the peak intensity ratios and R_2 changes are critical to our interpretation. Furthermore, severe overlap in the NMR spectra of these two constructs (because of the many repeated amino-acid motifs) would have made assignment difficult. Overlapping peaks also provide limited residue-specific information. In these constructs, we therefore only analyzed the well-resolved peaks to avoid bias due to peak overlap..... ” in the revised manuscript.

2. The data shown in Figure 2 suggests the potential to develop a true phase diagram, rather than the cartoon presented in Figure 2E. If feasible, the authors should consider quantifying the monomer vs phase separated fractions as a function of temp/salt/protein, to provide a more quantitative analysis in this panel.

We thank the reviewer for this suggestion. A more detailed analysis of various salt, protein, and temperature conditions is shown in the revised Figs 2, S1 and S2. We have also investigated the phase behavior of the full-length protein in more detail (see our replies to Reviewer 2).

REVIEWERS' COMMENTS:

Reviewer #1 (Remarks to the Author):

The authors have done a great job to address my concerns. Nice work!

Reviewer #2 (Remarks to the Author):

I would like to commend the authors on a compelling set of revisions, and the rigorous and absolute engagement of all reviewer comments. The revised version has removed virtually all of the experiments I found concerning and has provided appropriate caveats for why operating at such high concentrations are meaningful. The discussion regarding the idea that phase separation offers an alternative means for multimerization (whereas other galactins have conventional folded domains and are multimeric) is extremely strong. In short, I am happy with the revised version and strongly support publication. I have a few minor comments for the authors consideration.

Minor comments:

- Abstract does not define 'agglutinate', so for someone outside the field it becomes hard to know what is being described
- Our lipopolysaccharide (LPS) micelle model shows that the NTDs form multiple weak interactions to other galectin-3 and aggregate LPS micelles
 - o Not sure what this means? Should this read "... to other galectin-3 molecules and the aggregated LPS micelles"
- "The coexistence curve could be determined using a temperature ramp search (for example, nucleation begins at temperatures above 19 °C under these buffer and protein concentration conditions; Fig. 2a)."
 - o I don't think it's necessary to state things that could have been done. I would have liked to have seen this actually done (!) but given it's not, I don't think the authors need to explicitly spell this out. Many things could have been done!
- On page 6 where aromatic residues are mentioned as being important, this should cite seminal work by Nott et al with DDX4 (2015 Molecular Cell paper), as this was the first place the aromatic residues were titrated in IDRs to modulate phase behaviour.
- When comparing 40 μM vs 400 μM – does the protein phase separate at 400 μM ? I think not, but this should be explicitly stated to avoid any confusion.
- Page 6 line 178 says "higher order oligomers" form. Is there any evidence for higher order oligomers? This may be true, but in classical LLPS of homopolymers the dilute phase should be monomer and the dense phase a percolated network – i.e. oligomers do not arise. While it may well be that in this system oligomers do form, if the authors make this claim they should have evidence.
- Page 7 line 190 – did not see orange arrow in figure?
- Page 8 line 206 – when speaking about aromatic residues the manuscript states that aromatic residues reduce the propensity to phase separate. I suspect this should read that loss of aromatic residues reduces propensity to phase separate!
- Page 8 line 207 – they reduce their predicted propensity to form pi-pi (must include word predicted, as this is purely a predictive metric not direct measurement).
- Page 8 line 218 – remove word 'rough' (colloquial)
- Page 10 line 284 – the authors make reference to "stickers", which I suspect comes from the stickers and spacers model of Pappu et al. I would suggest citing the Wang et al 2018 Cell paper when stickers are invoked here, as it provides the context for thinking about stickers and explains what stickers are.

REVIEWERS' COMMENTS:

Reviewer #2 (Remarks to the Author):

I would like to commend the authors on a compelling set of revisions, and the rigorous and absolute engagement of all reviewer comments. The revised version has removed virtually all of the experiments I found concerning and has provided appropriate caveats for why operating at such high concentrations are meaningful. The discussion regarding the idea that phase separation offers an alternative means for multimerization (whereas other galactins have conventional folded domains and are multimeric) is extremely strong. In short, I am happy with the revised version and strongly support publication. I have a few minor comments for the authors consideration.

We thank the reviewer for carefully reading through our revised manuscript.

Minor comments:

- Abstract does not define 'agglutinate', so for someone outside the field it becomes hard to know what is being described

A short phrase "(acting as a "bridge" to aggregate glycosylated molecules)" has been added to explain this verb in the abstract.

- Our lipopolysaccharide (LPS) micelle model shows that the NTDs form multiple weak interactions to other galectin-3 and aggregate LPS micelles

- Not sure what this means? Should this read "... to other galectin-3 molecules and the aggregated LPS micelles"

We have added the word "then" to make this sentence clearer. "Our lipopolysaccharide (LPS) micelle model shows that the NTDs form multiple weak interactions to other galectin-3 and then aggregate LPS micelles"

- "The coexistence curve could be determined using a temperature ramp search (for example, nucleation begins at temperatures above 19 °C under these buffer and protein concentration conditions; Fig. 2a)."

- I don't think it's necessary to state things that could have been done. I would have liked to have seen this actually done (!) but given it's not, I don't think the authors need to explicitly spell this out. Many things could have been done!

I agree. The sentence "for example, nucleation begins at temperatures above 19 °C under these buffer and protein concentration conditions" has been removed.

- On page 6 where aromatic residues are mentioned as being important, this should cite seminal work by Nott et al with DDX4 (2015 Molecular

Cell paper), as this was the first place the aromatic residues were titrated in IDRs to modulate phase behaviour.

I am surprised that we did not cite this article. It inspires many of our studies (including another protein TDP-43). We have followed many protocols from this article (for example, using the thermostatic stage to monitor the phase transition). This article does not fit well into the content the reviewer mentioned, but we have cited this article in the Introduction in the revised version to emphasize its importance.

- When comparing 40 μM vs 400 μM – does the protein phase separate at 400 μM ? I think not, but this should be explicitly stated to avoid any confusion.

We add “(in the single-phase regime)” to avoid confusion (line 145 page 6).

- Page 6 line 178 says “higher order oligomers” form. Is there any evidence for higher order oligomers? This may be true, but in classical LLPS of homopolymers the dilute phase should be monomer and the dense phase a percolated network – i.e. oligomers do not arise. While it may well be that in this system oligomers do form, if the authors make this claim they should have evidence.

We thank for pointing this out. In that sentence, “into higher-order oligomers” is removed to avoid ambiguity. The revised sentence is “These results indicate that NTD monomers assemble through intermolecular interactions involving aromatic residues.”

- Page 7 line 190 – did not see orange arrow in figure?

They are in the inserts of Fig. 4 for indicating concentration-dependent chemical shift perturbation (red and green NMR signals). They are now enlarged for clarity.

- Page 8 line 206 – when speaking about aromatic residues the manuscript states that aromatic residues reduce the propensity to phase separate. I suspect this should read that loss of aromatic residues reduces propensity to phase separate!

Thanks for pointing this out. This sentence has been revised as “.....the loss of them reduces their predicted propensity to form pi-pi interactions.”

- Page 8 line 207 – they reduce their predicted propensity to form pi-pi (must include word predicted, as this is purely a predictive metric not direct measurement).

The word “predicted” has been added.

- Page 8 line 218 – remove word ‘rough’ (colloquial)

It has been removed.

- Page 10 line 284 – the authors make reference to “stickers”, which I suspect comes from the stickers and spacers model of Pappu et al. I would suggest citing the Wang et al 2018 Cell paper when stickers are invoked here, as it provides the context for thinking about stickers and explains what stickers are.

This article has been added to the Reference. We also cite Holehouse and Pappu’s review in Biochemistry (2018), in which I’ve learned the phrase “sticker”. A figure in this Biochem article also clearly illustrates the sticker/spacer idea.